# Sustainable fishing harvest rates for fluctuating fish and invertebrate stocks

**Alicia Poot-Salazar**[1◉], **Iván Velázquez-Abunader**[2◉]*, **Otilio Avendaño**[3‡],
**Polo Barajas-Girón**[4‡], **Ramon Isaac Rojas-González**[1◉], **Saul Pensamiento-Villarauz**[1‡], **Jesús M. Soto-Vázquez**[1‡], **José F. Chávez-Villegas**[1‡], **Rubén H. Roa-Ureta**[5◉]*

**1** Instituto Mexicano de Investigación en Pesca y Acuacultura Sustentables (IMIPAS), Mérida, Yucatán, México, **2** Centro de Investigación y de Estudios Avanzados del Instituto Politécnico Nacional (CINVESTAV), Mérida, Yucatán, México, **3** Unidad Multidisciplinaria de Investigación, Facultad de Ciencias, Universidad Nacional Autónoma de México (UNAM), Sisal, Yucatán, México, **4** COBI, Comunidad y Biodiversidad, Mérida, Yucatán, México, **5** Independent Consultant in Statistical Modelling, Marine Ecology and Fisheries, Portugalete, Bizkaia, Spain

◉ These authors contributed equally to this work.
‡ OA, PBG, SPV, JMSV, and JFCV also contributed equally to this work.
* jvelazquez@cinvestav.mx (IVA); ruben.roa.ureta@protonmail.com (RHRU)

**Data Availability Statement:** The data underlying the results presented in the study are available from Alicia Poot-Salazar, alicia.poot@imipas.gob.

## Abstract

Ecological theory predicts fluctuations, such as oscillations and instabilities, in populations whose dynamics can be represented by discrete-time surplus production models, whenever the intrinsic rate of population growth ($r$) is too high. Many fished stocks may have sufficiently high $r$ to undergo fluctuations under fishing. The maximum sustainable yield (MSY) is the fishing harvest rate concept that underlies United Nations Sustainable Development Goals and much of national fisheries administration around the world and yet in fluctuating stocks the MSY does not exist. This is because MSY's existence necessitates stable zero growth rates and in fluctuating stocks the growth rate switches from positive to negative over regular or irregular cycles, never staying put at zero. A more general surplus production concept is the total latent productivity (TLP). TLP averaged over years of negative and positive productivity has been proposed as a sustainable annual harvest rate for fluctuating stocks. We tested this theory assessing two harvested octopus populations inhabiting the Yucatan Peninsula with a 22-years time series of data, and a two-stages stock assessment methodology, with time-varying parameters at both stages. We find that parameters of the population dynamics changed in both species, dividing the time series in two periods, leading from single-point equilibrium to fluctuating dynamics in one species and increased amplitude and amplitude variability in the other species. These results mean that management based on the MSY would lead to overfishing and collapse of the two octopus stocks, as shown by stochastic projections. Conversely, the average TLP yielded much lower and realistic annual harvest rates, closer to actual landings over the 22-years period. We conclude that average TLP is the correct sustainable harvest rates for fluctuating stocks, which may include cephalopods, other invertebrates and small pelagic fish. This more general concept of surplus production needs to be incorporated in

mx in https://drive.proton.me/urls/
PXETCP7AP4#dMnHHCWeApXY.

**Funding:** Family Foundation, grant 2018-524 to the
Sustainable Fisheries Partnership. Walton Family
Foundation, grant 00104754; Lucile Packard
Foundation, grant 2021-73213; Sandler
Foundation, grant 2022-0429; Marisla Foundation,
grant 2022-0131; to Comunidad y Biodiversidad, A.
C. (COBI).

**Competing interests:** The authors have declared
that no competing interests exist.

multilateral and national fisheries management policies to avoid overfishing stocks that
have fluctuating population dynamics.

## Introduction

Despite a long history of criticisms, the maximum sustainable yield (MSY) remains in practice
a key concept in the generation of biological reference points to harvest fish stocks sustainably.
Originally derived from surplus production models and currently applied more generally [1–
3], the MSY is in policy documents of Sustainable Development Goal target 14.4 of the United
Nations, as well as in legislation at the national [4] and regional levels [5]. Originating in classi-
cal ecological theory [6], the MSY currently is not generally considered as a desired harvest
rate but rather, a generator of limit reference points, namely the biomass that produces the
MSY and the fishing mortality exerted on the stock by taking the MSY [7].

Discrete-time surplus production models are widely used in stock assessment. Ecological
theory predicts oscillations and instabilities in populations whose dynamics can be described
by these models [8]. Oscillations and instabilities are different types of variability, the former
generally describing regular cycles (with the exception of damped oscillations) and the latter
irregular variability, so we refer to both of these concepts as 'fluctuations'. These fluctuations
are triggered when the intrinsic rate of population growth ($r$) is sufficiently high. Starting as
peaks and troughs in biomass that repeat more or less regularly when $r$ is moderately high, it
progresses to irregular and chaotic variability as $r$ increases further [8, 9]. Many large and
small marine populations that are exploited by fishers can have high $r$ due to their life history.
A short-life cycle is connected to high $r$ [10, 11], for instance in octopus [12], and small pelagic
fish [13], but even long-lived stocks may have high $r$ [14–17] possibly when fecundity is high.
Furthermore, an empirical relationship in ecology spanning from viruses to whales shows that
given the body size of most fish and invertebrates harvested by fishers, $r$ may well fall in the
range of less than one to as high as ten per year [6, 17]. In simple discrete-time surplus produc-
tion models of the Graham-Schaefer type, May [8] found that fluctuations started to appear for
$r > 2.526$ per year but in more general formulations such as Pella-Tomlinson's, fluctuating
behaviour may start at lower $r$.

Fluctuating population dynamics could be common in fished stocks that lack formal assess-
ment. Most of un-assessed stocks are from tropical and sub-tropical regions [18], exploited by
small-scale, artisanal fleets in coastal areas and many are short-lived, falling on the fast extreme
of the slow-fast continuum in life history types [19]. Fluctuating stocks are characterized by
having an unstable equilibrium point. For instance, under a Shepherd stock-recruitment (SR)
relationship, the equilibrium point, located at the cross of the diagonal of equal numbers of
recruits and spawners and the SR curve, is unstable [20]. Any small disturbance from the equi-
librium point may bring the stock to fluctuations, either oscillating or irregular. Ricker-type
SR models can also generate fluctuations when the slope at the equilibrium point is steep and
spawning is concentrated in older animals [21]. Communities that contain several fluctuating
populations that are asynchronous in their cycles may contribute to ecosystem stability by
averaging out their cycles, so that in the aggregate variability decreases [13, 22, 23], and over
evolutionary time scales diversity increases [24].

In fluctuating stocks, the conditions are not present for the existence of the MSY because
the population growth rate never takes the zero value, switching instead from positive to nega-
tive at regular or irregular intervals. Surplus production is followed by deficit production in a

persistent cycle. Those fluctuating stocks in which surpluses generally exceed deficits are viable, in the sense that the mid- and long-term population growth rate is non-negative. The non-existence of the MSY in fluctuating stocks raises questions for fisheries management that are currently not addressed and that may become more important as more small-scale fisheries are assessed. This is because these fisheries usually harvest short-lived stocks and coastal stocks that may be more susceptible to enter into fluctuations. Thus two questions arise: in fluctuating stocks (i) what is the limit sustainable harvest rate and (ii) what to use as reference point for the generation of lower limits of stock biomass and upper limits of fishing mortality? In stable stocks the limit sustainable harvest rate is the MSY, and the reference points are the biomass and the fishing mortality at the MSY. For fluctuating stocks, Roa-Ureta et al. [16, 20] proposed to use the total latent productivity (TLP), averaged over positive and negative growth rate periods, as limit harvest rate. Latent productivity is the general biomass production concept [25] of which the MSY is the particular case corresponding to those stocks that can maintain their growth rate constant. Because unlike the MSY, the TLP is not a constant, varying year to year following the fluctuations of stock biomass, it was proposed as a novel solution to the well-known challenges that fluctuating stocks like small pelagic fish present when trying to determine their status with reference to the MSY [13]. In this work we tested the theory of fluctuating stocks, and the proposal to use the average TLP as a sustainable limit harvest rate, with a case study of two species of octopus that support the largest octopus fishery on the American continent, taking place in the continental shelf off the Yucatan Peninsula, Mexico.

Octopus stocks are interesting case studies of fluctuating dynamics and its consequences in fishery management. Landings records of *Octopus vulgaris* in West Africa and southern Spain show wide fluctuations that have been connected to environmental factors [26–29]. In northern Spain, fluctuating dynamics appears as a result of the spawning stock and recruitment relationship possessing an unstable equilibrium point [12, 20]. Doubleday *et al*. [30] have presented the hypothesis that cephalopods in general and octopus in particular are proliferating in the oceans due to changing environmental conditions, and that this proliferation is reflected in higher cephalopod catches worldwide. Our case study, the Yucatan octopus fishery, is the third largest octopus fishery in the world [31], and is based on two species: *Octopus maya* and *Octopus americanus*, the latter recently identified as a new species [32]. Landings of both species reach up to tens of thousands tonnes, comparable to landings from Western Africa and China, and landings of *O. maya* have been increasing in recent years. In this work we analyse a database of weekly landings, fishing effort, and mean weight of octopus in the catch over a period of 22 years. With these data we test the hypothesis of the utility of the average TLP to estimate sustainable harvest rates in stocks with fluctuating dynamics, using a stock assessment methodology that include time-varying dynamics in both species. The test is conducted by projecting the stock ten years into the future to show that management based on the MSY would lead to collapse of the stocks while management based on the average TLP would lead to sustainable exploitation.

## Materials and methods

### Description of the study area and the fishery

The study area was the Yucatan Peninsula, located in the eastern part of Campeche Bank, in the southeast of the Gulf of Mexico (Fig 1). Its continental shelf is subject to the influence of strong marine currents originating in the Loop Current, an important inflow of water from the Caribbean [33]. The incidence of the Loop Current provides cold and nutrient-rich oceanic water to the continental shelf [34]. Thus the eastern part of the Campeche Bank is one of the most important upwelling areas of the Tropical Atlantic [34]. The dynamic uplift

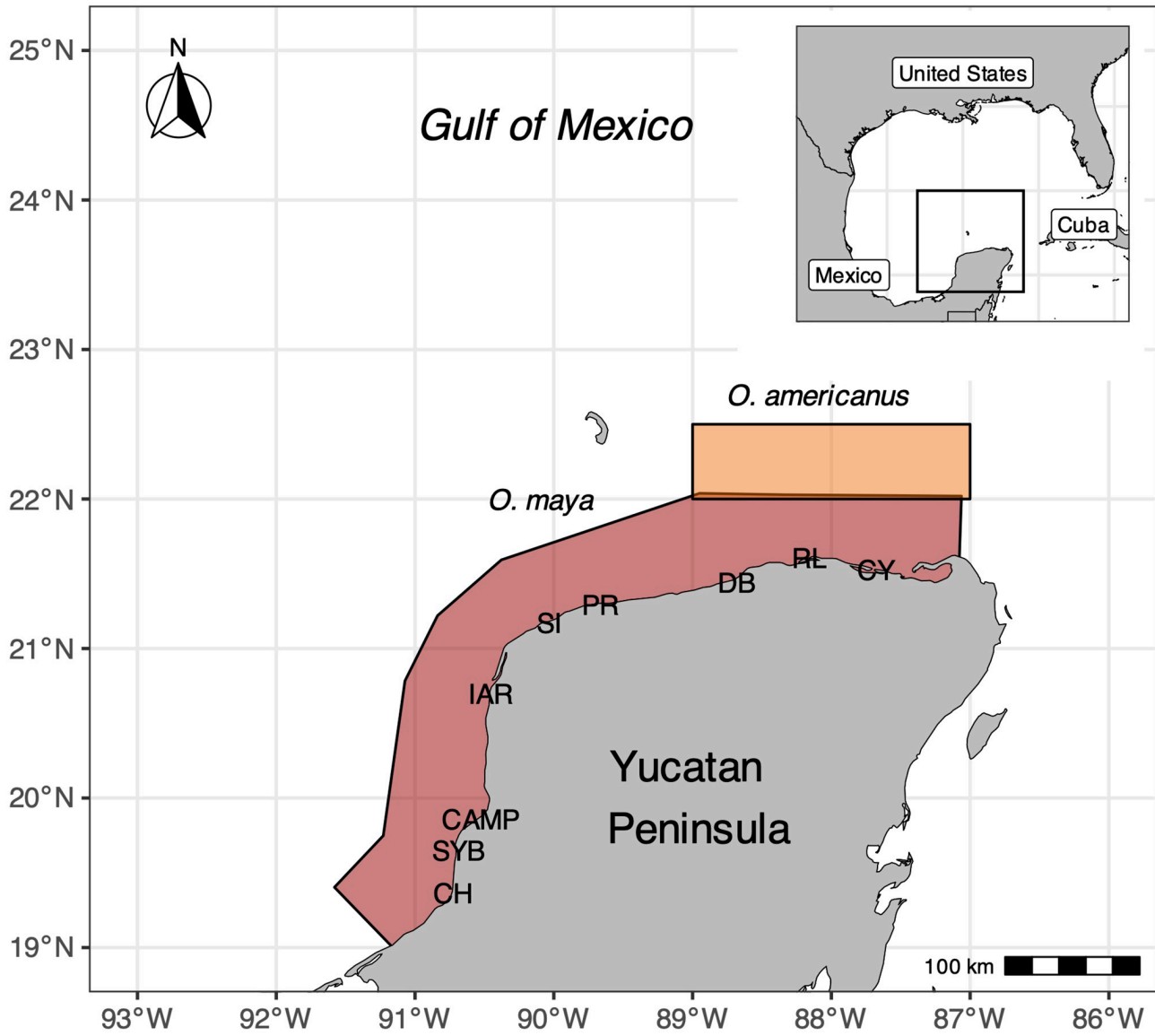

**Fig 1. The study area of the application examples.** The Yucatan Peninsula, fishing grounds of the Mexican octopus fishery, and main communities involved in octopus fishing (CH, Champotón; SYB, Seybaplaya; CAMP, Campeche; IAR, Isla Arena; SI, Sisal; PR, Progreso; DB, Dzilam de Bravo; RL, Río Lagartos; CY, El Cuyo). Colored shaded areas and adjacent continental shelf correspond to the Campeche Bank.

associated with the Loop Current develops seasonal upwellings in spring and summer [35]. Later, from October to March (autumn and winter), the recurrence of cold and dry air masses from the north of the continent produces strong frontogenesis in the area (gusts > 100 km/h), locally known as "nortes", that mix the water column on the continental shelf homogenizing oceanographic conditions [33, 36].

In the Yucatan Peninsula, octopus species have been fished since the ancient Mayans, while commercial operations began in 1949 in Campeche, and in 1970 in Yucatan and Quintana Roo [37]. The main species is the mayan octopus (*Octopus maya*), an endemic coastal species. With governmental support to renew the fleets and improve their navigation equipment around 1995, American octopus (*Octopus americanus*) landings from offshore fishing grounds

came to represent approximately 28% of the aggregate landings of the two species in the first decade of the 21st century. In the last two decades, both species together have recorded landings of over 20,000 tonnes. Aggregate landings have been increasing since 2011. *O. americanus* is still recorded in official landings as *O. vulgaris*, though genetic results [32] support the identification of *O. americanus* (formerly known as *O. vulgaris* type II) as the species targeted in the Yucatan Peninsula.

The Mexican octopus fishery in the area of the Yucatan peninsula is a small-scale artisanal fishery that operates during a season that runs from August to December of each year. The permitted fishing gear is called *gareteo* or *al garete* which would translate to drift rods [31, 38], a particular system of multiple fishing lines (numbering 4 to 16) that are tied to bamboo or wooden poles fixed to the boat. The lines are drifting for four to eight hours, using crabs or fish heads to attract the octopus. Two segments of the fleets operate in different depth ranges. One fleet segment comprises small boats of 5 to 10.5 m in length equipped with an outboard motor, and carrying in its interior one or two smaller boats called *alijos* [31]. The alijos, 2.5–3 m in length each, actually conduct the fishing drifting close to the main boat. This coastal segment of small-sized boats conducts one-day fishing trips and operates in shallow waters down to 20 m depth. A second fleet segment is composed of midsize boats with 12 to 25 m in length and it operates in areas deeper than 20 m staying at sea for extended periods, up to over a week with fishing days of up to 11 hours. Each of these midsize boats works as a mother vessel [38] carrying from 4 to 12 *alijos*. The small boats (coastal) fleet comprise approximately eight thousand boats and the midsize boats (offshore) fleet nearly five hundred boats.

The fishing grounds of each species are not mutually exclusive, and differ to some extent. *O. maya* is fished in shallower waters (<20 m depth) of an extended area bordering Yucatan, Campeche, and Quintana Roo states (Fig 1). Small and midsize boats operate in this coastal *O. maya* fishery. *O. americanus* has been traditionally fished in a rather small area (5,000 km$^2$), at the northern tip of the Yucatan Peninsula, in deeper waters of the extended continental shelf off Yucatan (Fig 1) by midsize boats only [31]. Nevertheless, recent investigations have shown that the two species occupy the eastern area of the Yucatan Peninsula in similar abundances [39, 40], Hence, small boats operating in that area are catching *O. americanus* some of which might be misclassified as *O. maya*.

Government regulation of the fishery consists of legal fishing only for registered fishers that have received fishing licenses, a minimum legal size of capture of 110 mm for mantle length or approximately 450 g for whole body weight, gear restrictions and seasonal closure (January to July). This management has been considered adequate [41] despite the fishing season operating in part of the growing and reproductive period of *O. maya* [42]. Fifty eight fishing-license holders were operating on *O. maya* and 192 on *O. americanus* in the first season of our time series, in 2000, and these numbers remained fairly constant during the first decade. Since 2010 and up to 2021 the average number of license-holders increased to 123 in the *O. maya* fishery and to 277 in *O. americanus* fishery.

## Description of the database

A national fisheries database from fishing vessels less than 10 tonnes in weight (small-scale or artisanal boats) called *Avisos de Arribo* (landing notices) was made available for stock assessment. This database covers the whole fishery and is organised by fishing trip, so it is a database with fine-grain time resolution. We compared total annual landings from *Avisos de Arribo* database to the annual total landings reported to the Food and Agriculture Organization (FAO) in FishStatJ v. 4.01.5 database and the two-time series for each octopus species were nearly identical. The *Avisos de Arribo* database contains data on boat identification, location of

fishing by name of area, species identification, weight of catch in kg, and fishing effort in effective days. This is a measure that counts the total number of fishing days of all *alijos* that are carried to the fishing grounds by each small and midsize boat. Thus, the measure of effort itself contains complete information on the effect of more *alijos* being carried by midsize boats compared to small boats. So, given that (a) the fishing effort data contains complete information on the different fishing powers of small and midsize boats on account of carrying several *alijos*, and (b) that small and midsize boats use the same fishing technique, we aggregated fishing effort and catch data for stock assessment purposes into a single fleet, a fleet of *alijos*. The database spans the 22-year period starting on 2000–01–01 and ending on 2021–12–31. It contains a total of 147,311 fishing trips for *O. maya* and 129,016 fishing trips for *O. americanus*.

In light of stock assessment results, an additional database of VMS location data was made available to us to examine area coverage by the fleet over time. This database contained records of locations of fishing operations from 2007 to 2022. We divided the database into four periods: 2007 to 2010 ($n = 2, 433, 415$), 2011–2014 ($n = 4, 195, 153$), 2015–2018 ($n = 4, 508, 094$), and 2019 to 2022 ($n = 1, 211, 048$) to better illustrate the process of expansion that characterized the evolution of the fishery during the study period.

Depletion stock assessment models (see below) run at discrete time steps, either day, week or month, so the fishery trip-to-trip database had to be aggregated over a time step. Most of the fishing trips lasted less than one week in the operations on both species over the time series (89% in the operation on *O. maya*, 79% in the operations on *O. americanus*). Therefore, we aggregated the fishing effort and catch data at the weekly time step in each of the 22 seasons, with the week enumerated at the time of sailing. Since the seasons start in August and end in December each year, the weekly time step produced between 19 and 23 weeks of data for each season.

The aggregation of raw data over weekly time steps incurred an error of time attribution for some fishing trips that lasted more than one week. These fishing trips were attributed to the first week of fishing even though they spanned two consecutive weeks of work at sea, and then included in the assessments. This error of time attribution could have been avoided by aggregating the raw data at the monthly time step, because all fishing trips lasted less than one month. However, at monthly time steps it is not possible to analyze each season separately with depletion models, because each season only covers six months. Therefore, we judged that the gain obtained from analyzing the fast weekly fishing dynamics, instead of the slower monthly fishing dynamics, was more important than the loss incurred by modest errors in time attribution. At the weekly time step the depletion stock assessment method employed in this work allows direct estimation of annually-varying fishery and population dynamics parameters and better captures the fast growth and high natural mortality of octopus life history.

Stock assessment as implemented in this work requires turning the catch in weight per week into catch in numbers per week, thus needing a parallel time series of mean individual octopus weight in the catch. We supplemented the total effort and catch database with samples of biological data collected by regular programs of IMIPAS, the Mexican national fisheries research center. Additional data was recovered from specific scientific projects [39, 40]. These biological data did not cover the first nine years of the *Avisos de Arribo* database (2000 to 2008) in both octopus species, making it necessary to replace the missing data with predictions from accessory models. To this end, we implemented the methodology described by Roa-Ureta [43] that essentially replaces raw data and missing data with randomized predictions of mean weight through all time steps in a fishing season. It consists firstly of fitting a cubic spline model with function *loess* of the R statistical programming system [44] to the relation between mean weight in the catch and week with all available data, regardless of year of collection. The spline model produces both the expected mean weight per week for all weeks of the season and

the standard error of prediction of a typical season from a very large sample size for *O. maya* ($n = 87, 080$) and fairly large sample size for *O. americanus* ($n = 1, 669$). Secondly, to account for the error of prediction of the mean weight per week, we generated new random values from truncated normal distributions defined by the expected mean weight per week, its standard error as standard deviation, and a band of two standard deviations around the expected mean weight, using R package Runuran [45]. This resampling produces a randomized-predicted mean weight in the catch for every week of the time series. In this manner a complete database of fishing effort, catch in weight, and mean weight in the catch per week, was available for stock assessment for both octopus species, spanning August 2000 to December 2021, a total of 22 fishing seasons.

## Stock assessment

We employed the hierarchical statistical model developed in Roa-Ureta *et al.* [16]. This is a data-intermediate method requiring weekly time series of (1) total fishing effort in any unit, (2) total catch in weight, and (3) mean weight in the catch, in a time series of many years. This method consists of two stages: fitting multiple variants of intra-annual generalized depletion (IAGD) models by maximum likelihood [46] and then using estimates of annual biomass and their covariance matrix from the best IAGD variant to fit generalized surplus production models, also by maximum likelihood. The first stage of this methodology is relatively new [46]. It has been tested by comparing its predictions to independent survey data [47], by fitting it against simulated data in Monte Carlo simulations [48], and by comparing its predictions with those from well-established conventional methodologies [49]. It has been applied extensively since its introduction in a wide variety of contexts [15, 16, 20, 43, 48–56]. A schematic representation of the methodology is in Fig 2 of Roa-Ureta *et al.* [54].

In the first phase we fitted IAGD models to the weekly catch, effort and mean weight data of each season and each octopus species separately, to generate maximum likelihood estimates of total stock biomass at the start of each season (i.e end of July each year). These IAGD models also produced estimates of the average natural mortality experienced by the stock in each season, abundance inputs such as recruitment of octopus that grow to the size ranges retained by fishers as well as fleet area expansion that add stock to the vulnerable biomass, and several fishing operation parameters. Since each season is analyzed separately, all parameters in the models are allowed to vary season to season, thus possibly discovering annually-varying parameters in octopus population dynamics and fishing techniques and tactics.

In the second phase we used biomass predictions at the start of each season from IAGD models to fit Pella-Tomlinson surplus production models to the annual biomass of each octopus species. At this stage, we passed the statistical uncertainty in biomass predictions from IAGD models to the inference on parameter values of the surplus production model by implementing a non-Bayesian hierarchical inference technique [16, 57]. This consists in maximizing a self-weighting marginal-estimated likelihood function built upon the multivariate normal distribution, whose mean is the vector of predicted annual biomass and whose variance-covariance matrix is built using the standard errors of the annual biomass estimates. The variance-covariance matrix in this multivariate likelihood is symmetric and diagonal by design, because each biomass estimate is obtained from separate fits of depletion models to each season's data, so biomass estimates are uncorrelated. At this stage we also tested time-varying population dynamics parameters and contrasted those with a null model of conventional constant-parameter population dynamics.

**Intra-annual generalized depletion (IAGD) models.** This part of the methodology follows the advice in Arkhipkin *et al.* [58] to assess cephalopods stocks using innovative depletion

models. IAGD models analyze the fast daily or weekly fishing dynamics inside an annual fishing season to produce maximum likelihood estimates of stock's total abundance at the start of the season ($N_0$), mean natural mortality over the season ($M$), a generalized catchability coefficient (the scaling, $k$), episodic in-season inputs and outputs of abundance ($R$ for abundance inputs and $S$ for emigration outputs), a power modulator of the effect of fishing effort on catch ($\alpha$) and a power modulator of the effect of abundance on catch ($\beta$). Unlike the previous application to an octopus fishery in northern Spain [20], in this case the fishing season does not include the main spawning season of any of the two species. These happen in February to June according to the most recent findings about *O. maya* reproduction [59] and in January to July in *O. americanus* (authors' unpublished results from Project 237057, CONCACYT, Mexico). Therefore, IAGD models in this application did not include any spawning emigration pulse ($S = 0$ for all time steps in all seasons). Under these definitions, IAGD models applied to each season's data and each octopus species separately were of the form

$$
\begin{aligned}
C_t \quad &= kE_t^\alpha N_t^\beta = kE_t^\alpha m f_t(M, N_0, C_{i<t}, R, S) \\
&= kE_t^\alpha m \left( N_0 e^{-Mt} - m \left[ \sum_{i=1}^{i=t-1} C_i e^{-M(t-i-1)} \right] + \sum_{j=1}^{j=u} I_j R_j e^{-M(t-\tau_j)} \right)^\beta
\end{aligned}
\tag{1}
$$

$$
t, k, N_0, \alpha, \beta, M, R_j > 0, C_t, E_t \geq 0
$$

where $C_t$ is the true and unobserved catch at week $t$, $E_t$ is the observed effort, $m = exp(-M/2)$ is an adjustment that makes all catch happen instantaneously at mid-week, $N$ is latent stock abundance, the index $j$ runs over a number $u$ of abundance inputs that occur at certain time steps $\tau_j$ and the indicator variable $I_j$ is zero before $\tau_j$ and is one at $\tau_j$ and after. In this model the set of parameters that are estimated simultaneously by likelihood maximization is $\boldsymbol{\theta}_{IAGD} = [k\ M\ N_0\ \{R_j\}\ \alpha\ \beta]$. All parameters were estimated simultaneously from good initial values and without any restriction.

The number of abundance inputs $u$ and their timings $\tau_j$ are non-differentiable parameters so they are estimated by fitting alternative hypotheses for $u$ and the $\tau_j$ and then selecting the best working model from objective criteria, including maximum likelihood. In this work we fitted models having from $j = 1$ to $j = u$ abundance inputs during each season in the data from both octopus species. The timing $\tau_j$ and maximum number $u$ of those inputs were determined in the following way: we started by fitting a pure depletion model, with no input of abundance along the season. These pure depletion models are nearly always poorly determined and estimate implausible high abundance but they are informative because of the manner in which they fail. Specifically, we noted the weeks at which the most positive residuals from the pure depletion model were timed. These residuals usually stand out in the fit of a pure depletion model and point to potential inputs of abundance due to recruitment or fleet area expansion happening at specific weeks. Next we fitted a model with one input of abundance timed at the week of the most positive residual of the pure depletion model. Examining the 1-input model we once again noted the weeks of standing-out positive residuals to select a second week to time a new input of abundance. This forward procedure was continued until no more standing-out positive residuals could be clearly determined. In some cases, for a given number of inputs $u$, we additionally fitted models having alternative $\tau_j$ because residuals were standing out similarly in two or three weeks. For instance, for a 1-input model we may have fitted one model with the input timed close to the start of a season and another model having the single input timed at mid season. Therefore in several cases we had more than one model fitted with the same number of inputs of abundance. These procedures for determining $u$ and $\tau_j$ for each

season and each octopus species led to having multiple IAGD model fits for model selection. Hereafter we use the term 'variants' to refer to these alternative fits to the data of each season and octopus species.

In addition to having variants arising from fits with different $u$ and $\tau_j$ for each season's data, we produced more variants by using any of four likelihood functions for likelihood maximization and three methods for numerical optimization (spg, CG, and Nelder-Mead). In defining these likelihood functions we considered that the true total catch time series $C_t$ was not observed and instead a random variable $\chi_t$ with realizations close to $C_t$ was observed, so essentially IAGD models assume observation error in the catch time series. The four likelihood functions were two approximated likelihoods, the adjusted-profile normal likelihood and the adjusted-profile lognormal likelihood [60], and two exact likelihoods, the normal and lognormal, see Table 2 in [54] for precise formulas. The two former are approximations that eliminate the dispersion parameter and the two latter estimate the dispersion parameter simultaneously with parameters in $\theta_{IAGD}$.

As a result of these alternative configurations, each season's data for each octopus species produced dozens of IAGD variants. We applied model selection methods to identify one variant with the best properties for each season and octopus species. We employed numerical, statistical, and information criteria for variant selection as follows. Firstly, according to the numerical criterion we eliminated all variants where the largest numerical gradient in absolute value was higher than 1. When this happens it means that some or all parameter estimates are suspected to be biased because they are not evaluated at the maximum likelihood, where the gradient should be close to zero. Eliminating fits of statistical models with relatively large maximum absolute gradient is common practice in stock assessment, and the value of 1 as upper limit for retention is a reasonable standard [61–64]. In our study this first filter eliminated just a few variants, usually none or just one. Secondly, we employed two statistical criteria. One was examination of the matrix of pairwise correlations between parameter estimates. In a good statistical model, these correlations should be clustered around zero because this means that each parameter in the model played a necessary role in explaining the data. The other was the coefficients of variation (CVs) of estimated parameters for each variant. The best model will have low CVs (ideally less than 50%) in all or most of the estimated parameters, especially parameters that are involved in the estimation of biomass at the start of the season ($M$, $N_0$ and the $\{R_j\}$) because the estimated biomass will be determining the fit of the surplus production model in phase two of the hierarchical inference framework. Thirdly, we examined the Akaike Information Criterion (AIC), and therefore the maximized log-likelihood, comparing variants with the same likelihood function. The three types of criteria were applied sequentially, so statistical criteria were applied to the variants remaining after application of the numerical criterion, and the information criterion was applied to the remaining variants after application of statistical criteria. This variant selection process did not introduce bias in parameter estimates because it was based on objective quality-assurance criteria. In fact the variant selection process removed biased variants because variants with poor definition of the number of abundance inputs $u$ tend to estimate very high abundance and natural mortality rates that do not accord with the known lifespan of octopus.

After selection of the best variant for each fishing season in each octopus species, we computed derived quantities of relevance to determine stock's status and to estimate the surplus production model in phase two. These were total abundance per week, fishing mortality per week and total biomass at the start of the season. Total abundance per week was calculated from the formula inside the large parentheses in Eq 1. Fishing mortality per week ($F_t$) was calculated solving for $F$ in Baranov's catch equation with the known values of $M$, $N_t$ and $C_t$. Solving for $F_t$ was done using R's *uniroot* function. Finally, total biomass at the start of the season

was calculated by adding $N_0$ to the back-calculated abundance from abundance inputs (the $\{R_j\}$) and then multiplying the resulting addition by the mean weight of octopus at the start of the season ($\bar{w}_0$),

$$\hat{N}_{0,Total} = \hat{N}_0 + \sum_{j=1}^{u} \hat{R}_j e^{\hat{M}t}, \quad \hat{B}_{0,Total} = \hat{N}_{0,Total}\bar{w}_0 \tag{2}$$

It is important to obtain a measure of statistical uncertainty of the $\hat{B}_{0,Total}$ estimate because the set of $\hat{B}_{0,Total}$ estimates across all seasons for each octopus species is used in phase two to estimate hyper-parameters of the surplus production model. This latter model must take into account that some $\hat{B}_{0,Total}$ from some seasons are more precise than other estimates in order to correctly propagate statistical uncertainty from raw data to the top model in the hierarchical inference, i.e. in order to build a self-weighting likelihood function [65, 66]. Therefore we calculated the standard error of each $\hat{B}_{0,Total}$ estimate using the delta method in time serial fashion using Eq 2. This method has been found to perform well and faster in stock assessment applications [67].

All procedures described in this sub-section, including data preparation, exploratory modelling to find good initial values for parameters before optimization, likelihood optimization, plotting of results, model selection and prediction of derived quantities as well as uncertainty calculations using the delta method, were carried out with functions in R package CatDyn v. 1.1–1 [68].

**Pella-Tomlinson surplus production models.** Initial annual biomass and their standard errors for each of the two octopus species over the 2000 to 2021 period as well as total annual landings were given as observations to fit a Pella-Tomlinson surplus production model with the following parameterization:

$$B_y = B_{y-1} + r_1 B_{y-1}\left(1 - \left(\frac{B_{y-1}}{K_1}\right)^{p_1-1}\right) - C_{y-1}, \quad 2000 \le y < y_x$$

$$B_y = B_{y-1} + r_2 B_{y-1}\left(1 - \left(\frac{B_{y-1}}{K_2}\right)^{p_2-1}\right) - C_{y-1}, \quad y_x \le y < 2021 \tag{3}$$

$$r_1 > 0, K_1 > 0, p_1 > 1, r_2 > 0, K_2 > 0, p_2 > 1$$

where $B_y$ (tonnes) is the biomass at the start of each year $y$, $r_1$ and $r_2$ (1/yr) are the intrinsic rate of growth over periods covering 2000 to $y_x - 1$ and $y_x$ to 2021, respectively, $K_1$ and $K_2$ (tonnes) are the carrying capacity of the environment in the same periods, $p_1$ and $p_2$ (dimensionless) are the shape of the production function of the same periods, and $C_y$ are the observed annual catch totals, assumed to have no observation error. Note that from a mathematical point of view the parameter controlling the rate of growth $r$ in Eq 3 is $r/(p - 1)$ [69] but this is trivial since $p$ is dimensionless. The two periods comprising 2000 to $y_x - 1$ and $y_x$ to 2021 are called here pre-expansion and post-expansion, respectively, for reasons that will become evident when examining the results of fitting IAGD models. The model in Eq 3 allows for either a single regime ($K$, $p$ and $r$ are the same for both periods) or two regimes of abundance ($K$, $p$ and/or $r$ differ for both periods). This simple approach to modelling time-varying parameters in surplus production models was also employed in Roa-Ureta *et al.* [52].

Time-varying dynamics was tested by setting model variants with changes from pre-expansion to post-expansion in one of the three parameters ($\boldsymbol{\theta}_{PTSP} = [K_1\,K_2\,p\,r]\vee[K\,p_1\,p_2\,r]\vee [K\,p\,r_1\,r_2]$), two of the three parameters ($\boldsymbol{\theta}_{PTSP} = [K_1\,K_2\,p_1\,p_2\,r]\vee[K_1\,K_2\,p\,r_1\,r_2]\vee[K\,p_1\,p_2\,r_1\,r_2]$),

or the three parameters together ($\boldsymbol{\theta}_{PTSP} = [K_1\ K_2\ p_1\ p_2\ r_1\ r_2]$). A model with no change in any of the three parameters from pre-expansion to post-expansion ($\boldsymbol{\theta}_{PTSP} = [K\ p\ r]$) was also fitted as a null model, thus totalling eight variants for each octopus species. In all variants, biomass just before the start of the annual time series (1999) was set as equal to $K_1$. Model selection was conducted using the same criteria as per selection of the best variant of the depletion models except that in this case the AIC played a more prominent role because all Pella-Tomlinson variants were fitted using the same likelihood function.

After model selection of the best variant for each octopus species we calculated the maximum sustainable yield (*MSY*),

$$MSY = rK(p-1)p^{\frac{-p}{p-1}} \tag{4}$$

the biomass that produces the MSY

$$B_{MSY} = Kp^{\frac{1}{1-p}} \tag{5}$$

and the total latent productivity (TLP) [25, see eq. 2.12 in]

$$\dot{TLP}_y = \gamma MSY \frac{B_y}{K}\left(1 - \left(\frac{B_y}{K}\right)^{p-1}\right) + C_y, \quad \gamma = \frac{p^{p/(p-1)}}{p-1} \tag{6}$$

These derived quantities were computed for the pre-expansion and post-expansion periods hypothesized in Eq 3. Standard errors were calculated with the delta method in R package Cat-Dyn [68].

To fit the model in Eq 3 to the time series of biomass observations from IAGD models and annual landings we maximized a marginal-estimated likelihood function

$$L_H(\{K_1, K_2, p_1, p_2, r_1, r_2\}|\{\hat{B}_y\}) \propto -\frac{1}{2}\sum_{y=2000}^{y=2021}\left(log(2\pi s_{\hat{B}_y}^2) + \frac{(\hat{B}_y - B_y)^2}{s_{\hat{B}_y}^2}\right) \tag{7}$$

where the $s_{\hat{B}_y}^2$ are the distinct standard deviations of each initial biomass estimate from IAGD models, $\hat{B}_y$ are the maximum likelihood estimates of initial biomass from IAGD models, and $B_y$ are the true initial biomass according to Eq 3. This likelihood function is self-weighting [65, 66] in the sense that it down-weights the effect of relatively imprecise estimates of $\hat{B}_y$ and up-weights the impact of relatively precise estimates inside the model when determining maximum likelihood estimates. This inference approach assumes multivariate observation error in biomass observations coming from the depletion model ($\hat{B}_y$).

The fit of Pella-Tomlinson model to IAGD biomass estimates and their standard errors was programmed in AD Model Builder using the ADMB-IDE user interface [70, 71]. Taking advantage of ADMB built-in functions we fitted the variants with bounded and unbounded parameters over maximization phases, and we produced annual biomass estimates and their standard errors using the *sdreport* function. When using bounded optimization, parameter estimates were not allowed to crash against any of the two bounds. ADMB output files were finally collected using the R programming system for calculation of derived quantities (Eqs 4, 5 and 6) and plotting.

To further show that the MSY is not adequate for fluctuating stocks such as the octopus species in Yucatan, we projected each stock using the Pella-Tomlinson model, parameters estimated here, and five scenarios of landings, ten years into the future, from 2022 to 2031. These scenarios included annual catch set at the level of the MSY, set at the level of the TLP, occurring at the level of the mean historical annual catch (2000 to 2021), and at 75% and 50% of the

mean historical annual catch. In these projections, we assumed that when harvesting set at the level of the MSY or the TLP, annual catches incurred a coefficient of variation of 25%, i.e. we assumed fairly wide implementation error. In the scenarios with catches at the level of mean historical catches, we assumed a coefficient of variation equal to the observed coefficient of variation (2000 to 2021). Likewise, in scenarios of catches at the 75% and 50% of mean historical catches we set the coefficient of variation at 75% and 50% of the historical coefficient of variation. We implemented re-sampling from truncated normal distributions with function *urnorm* in R package Runuran [45] to generate the landing time series, with one standard deviation to each side of the expected value as interval. The parameters of the Pella-Tomlinson model were collected from re-sampling over non-truncated normal distributions defined by a mean equal to the estimated value of the parameter and a standard deviation equal to the standard error of the estimated value of the parameter. Since we found that time-varying parameters Pella-Tomlinson models were better supported than time-invariant parameters models (see below), we used the estimated values for the later period to produce projections of stock's biomass. We replicated these projected trajectories 1000 times, saving both the landings and stock biomass trajectories. These algorithms were written in R 4.3.0. [44].

## Results

### Intra-annual generalized depletion (IAGD) models

Considering both octopus species, a total of 1,191 IAGD model variants were implemented and submitted to likelihood optimization, 912 of which reported successful convergence using three optimization methods available in R (S1 Table). Using variants selection criteria, 44 best variants were selected, 22 for each species, one for each season in each species. Forty one out those selected variants were fit with the adjusted profile normal likelihood, three with the adjusted profile lognormal likelihood, and just one with the exact normal likelihood. Likewise, 36 selected variants were fit with the CG method, five with the spg method, and three with the Nelder-Mead method.

The fit of the selected models to the data of each season and each species is shown in S1 and S2 Files, including the catch data by week and the model predicted catch, as well as residuals and quantile diagnostics plots. All model fits show good agreement between weekly catch data and model-predicted weekly catch (top panel, S1 and S2 Files), centred and symmetrical residual histograms (bottom-left panel), shapeless clouds of residuals (bottom-centre panel) and q-q plot dots falling on the 45° diagonal line (bottom-right panel).

All parameter estimates from IAGD models are shown in Fig 2. Natural mortality rate ($M$, annualized by multiplying the weekly estimate by 53) appears to be decreasing in *O. maya* and in general becoming more like $M$ in *O. americanus*, so from 2017 onward both species suffer natural mortality rates close to 1 per year. Some natural mortality estimates are very precise, especially in *O. maya* (2001, 2005, 2010, 2014, 2019, 2020, 2021) but also in *O. americanus* (2002, 2013, 2018). There is substantial annual variation in $M$ in both species, ranging from a minimum of 0.25 1/yr (2019) to a maximum of 6.4 1/yr (2005) in *O. maya* and 0.1 (2001) 1/yr to 2.8 1/yr (2005) in *O. americanus*.

Initial abundance (end of July each year) is always higher in *O. maya*, fluctuating around 100 million octopus, and it increased between 2015 and 2019, while initial abundance in *O. americanus* has been stable around 25 million octopus (Fig 2). Remarkably, initial abundance are generally estimated with excellent statistical precision. Adding over all abundance inputs (that vary from 1 to 5 in each species, see S1 Table) to obtain a single amount of additions to the vulnerable stock during each season, we observe similar magnitudes in both species (Fig 2). These additions contribute around 50 million octopus of each species as new

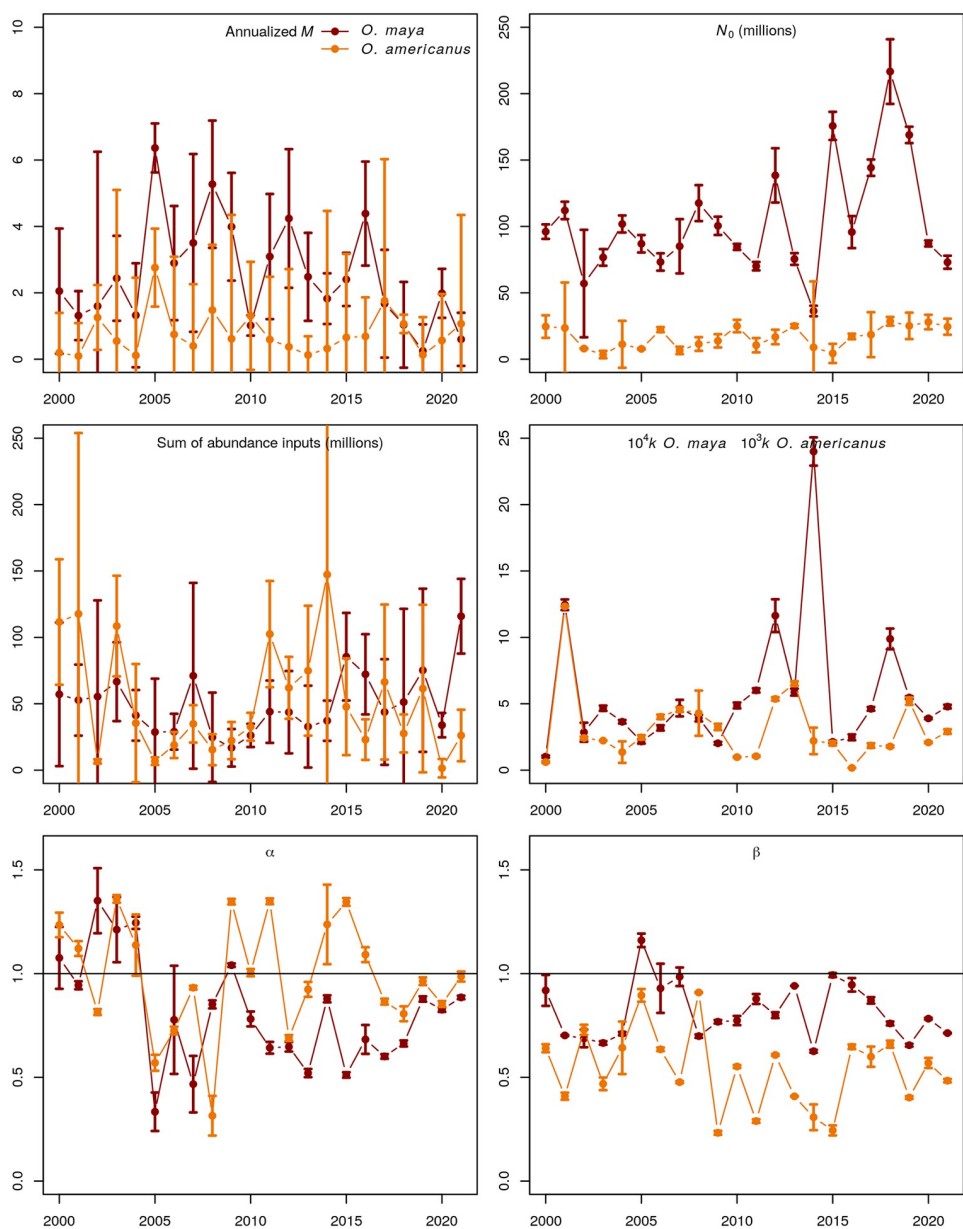

**Fig 2. IAGD estimates.** Maximum likelihood estimates of parameters and two standard errors bars of intra-annual generalized depletion models selected as best fits for the *O. maya* and *O. americanus* fisheries database in the Yucatan Peninsula, Mexico. a) Natural mortality rate, b) initial abundance, c) sum of abundance inputs during the season, d) scaling (note different multipliers for each species), e) effort response, f) abundance response.

vulnerable abundance during each season. The statistical precision of these estimates is generally acceptable though it is fairly poor in some years (*O. maya*: 2000, 2001, 2005, 2007, 2018; *O. americanus*: 2001, 2014, 2017, 2019).

The last three parameters presented in Fig 2, namely the scaling $k$, effort response $\alpha$, and abundance response $\beta$, are fishing operations parameters. The scaling shows that the capture of *O. americanus* has a generalized catchability one order of magnitude larger than the capture of *O. maya* and that in both species it may show substantial changes year to year, agreeing with previous results obtained with field data [72]. The fishing gear for both species is either

proportional ($\alpha = 1$) or saturable ($\alpha < 1$) and it appears to be turning more saturable for *O. maya* in recent years. Catch rates clearly are hyperstable in both species ($\beta < 1$) and more so in *O. americanus*. All three parameters of fishing operations are generally estimated with excellent statistical precision.

During the period of 2000 to 2016 the fleet generally exerted a greater instantaneous exploitation rate (F/[F+M]) on *O. americanus* than on *O. maya* with the exception of 2006, 2010 and 2013, when exploitation rates were similar in both species (Fig 3). In this earlier period, the exploitation of *O. maya* was generally conducted below the 40% threshold (Fig 3), a limit biological reference point recently used by the FAO for the management of a short-lived stock assessed using generalized depletion models [73]. However, exploitation rates on *O. americanus* generally exceeded 40%. This situation changed in 2017, with both stocks harvested close to 40% exploitation rate, except in 2019 when it exceeded 60% in both species (Fig 3).

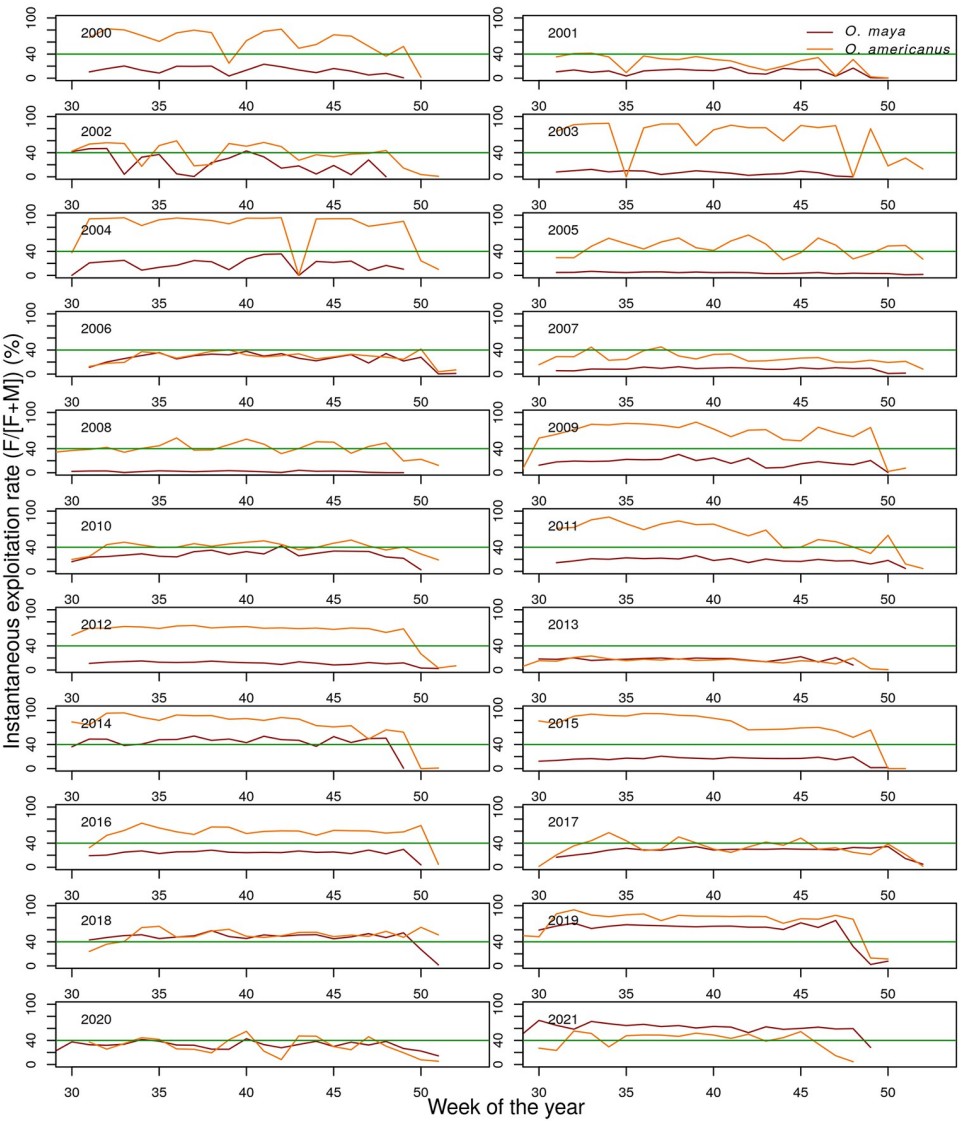

**Fig 3. Exploitation rates.** Weekly instantaneous exploitation rate ($100*F/(F + M)$) from intra-annual generalized depletion models applied to catch, effort and mean weight data in the fisheries for *O. maya* and *O. americanus* in the Yucatan Peninsula, Mexico. The thick black line marks the 40% exploitation rate.

Stock biomass at start of season as predicted by IAGD models shows a rather rich dynamics, with higher biomass in more recent years (Fig 4, green circles). Note that in Fig 4 the small circles are imprecise estimates of biomass from IAGD models so they have less influence in the biomass trajectory estimated by the surplus production models than biomass estimates

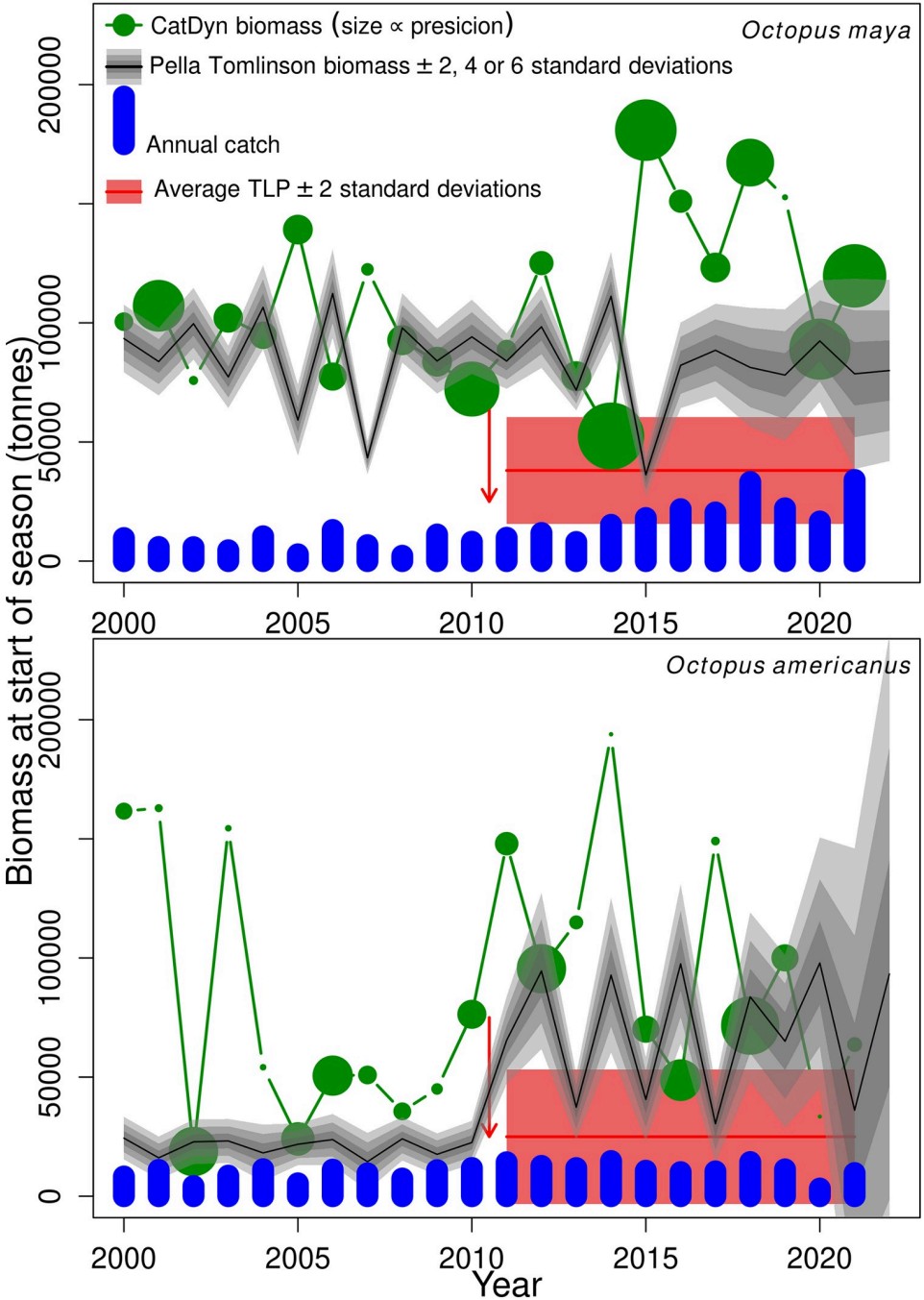

**Fig 4. Biomass dynamics.** Biomass dynamics (green dots and lines, black lines and surrounding bands) and realized (blue bars) and sustainable annual harvest rates (TLP: total latent productivity) of two octopus species in the Yucatan Peninsula, Mexico. The arrow points to the timing set to trigger changes in parameters of the biomass dynamics, the symmetry of the production function (*p*) in *O. maya* and the carrying capacity of the environment (*K*) in *O. americanus*. CatDyn biomass are weighted by $q(1/CV(Biomass))/(1/min(CV(Biomass)))$ where q = 0.3 and 0.15 in *O. maya* and *O. americanus* respectively, so more precise estimates appear bigger.

represented by large circles, by virtue of the self-weighting likelihood functions in the hierarchical inference framework. Examination of these initial biomass estimates indicate that the shift to higher biomass started to happen in both species from 2010 to 2011. Thus we set the transition year ($y_x$ in Eq 3) at 2011 in both species.

## Pella-Tomlinson surplus production models

In *O. maya*, the variant that had constant parameters over the 2000 to 2021 period, as well as variants with change in *r* and with changes in all three parameters (*K*, *p*, and *r*), had large gradients and AIC so these were discarded (Table 1). The variant with changes in *K* and *r* predicted a collapse of the stock in 2017 and the variant with change in *K* produced an unrealistic estimate of *r* ($> 23$ 1/yr) so these were also discarded. Of the remaining three variants (change in *p*, changes in *K* and *p*, and changes in *p* and *r*), the model with change in *p* had the best AIC (Table 1) and correlations between estimates (more clustered around zero) so the variant with a change in *p* in 2011 was selected as the best working model.

In *O. americanus*, the variants with change in *p*, change in *r*, changes in *K* and *p*, changes in *p* and *r*, and changes in all three parameters, had poor gradients and high AIC (Table 1) so these were discarded. The variant with constant parameters produced and estimate of *p* that was highly unrealistic ($> 7$) so it was also discarded. Of the remaining two variants (change in *K* and changes in *K* and *r*), the model with change in *K* had the lowest AIC (Table 1) and correlations more clustered around zero. Therefore in this species, the variant Pella-Tomlinson model with change in *K* was selected as the best working model.

In *O. maya*, biomass dynamics from the surplus production model showed fluctuating dynamics along the entire period, with decreasing amplitude as it approaches the switch year 2010 to 2011, wider amplitude in the subsequent six years, and decreased amplitude again at the end of the time series, all of these around a rather constant mean of close to 80 thousand tonnes (Fig 4). Conversely, in *O. americanus*, the biomass dynamics shows a clearly-defined jump to higher levels of biomass in the switch year of 2011 (Fig 4), a consequence of higher *K*

**Table 1. Model selection, surplus production model.** Comparison of surplus production model variants fitted to biomass and biomass standard errors predicted by intra-annual generalized depletion models applied to the fisheries for *O. maya* and *O. americanus* in the Yucatan Peninsula, Mexico. In both species changes in some parameter values happen from 2010 to 2011. Nº is the number of parameters. Best working model marked in bold.

| Species | Change in | Nº | Gradient | AIC | Anomalies |
|---|---|---|---|---|---|
| *O. maya* | | 3 | $> 100$ | -557 | $r > 23$ |
| | *K* | 4 | $< 0.01$ | -682 | |
| | **p** | **4** | **0.257** | **-686** | |
| | *r* | 4 | 23.621 | -660 | |
| | *K, p* | 5 | $< 0.01$ | -623 | Stock collapse in 2017 |
| | *K, r* | 5 | $< 0.01$ | -1136 | |
| | *p, r* | 5 | $< 0.01$ | -667 | |
| | *K, p, r* | 6 | $> 100$ | -593 | |
| *O. americanus* | | 3 | $< 0.01$ | -764 | $p > 7$ |
| | **K** | **4** | **$< 0.01$** | **-563** | |
| | *p* | 4 | $> 100$ | -1026 | |
| | *r* | 4 | $> 100$ | -675 | |
| | *K, p* | 5 | $> 100$ | -1037 | |
| | *K, r* | 5 | 1.569 | -551 | |
| | *p, r* | 5 | $> 100$ | -687 | |
| | *K, p, r* | 6 | $> 100$ | -539 | |

**Table 2. Parameter estimates, surplus production model.** Maximum likelihood estimates of directly estimated parameters ($K$, $p$, and $r$) and derived reference points (MSY, $B_{MSY}$, average total latent productivity (TLP)) of the best Pella-Tomlinson models for each octopus species, *O. maya* and *O. americanus*, in the Yucatan Peninsula, Mexico. MLE: maximum likelihood estimate; CV: coefficient of variation; TLP: annually averaged total latent productivity; $\bar{L}$: annually averaged landings.

| | *O. maya* | | *O. americanus* | |
|---|---|---|---|---|
| **Parameter** | **MLE** | **CV (%)** | **MLE** | **CV (%)** |
| $K_1$ (tonnes) (pre-expansion) | 93,416 | 2.5 | 24,379 | 6.1 |
| $K_2$ (tonnes) (post-expansion) | | | 82,354 | 4.8 |
| $p_1$ (pre-expansion) | 1.8408 | 0.8 | 1.9108 | 8.6 |
| $p_2$ (post-expansion) | 1.9557 | 1.1 | | |
| $r$ (1/yr) | 2.9687 | 2.1 | 3.5084 | 17.2 |
| $MSY_1$ (tonnes) (pre-expansion) | 61,304 | 3.1 | 20,025 | 2.7 |
| $MSY_2$ (tonnes) (post-expansion) | 67,174 | 2.4 | 67,646 | 8.1 |
| $B_{MSY,1}$ (tonnes) (pre-expansion) | 45,210 | 2.6 | 11,975 | 9.1 |
| $B_{MSY,2}$ (tonnes) (post-expansion) | 46,304 | 2.4 | 40,451 | 3.7 |
| $TLP_1$ (tonnes) (pre-expansion) | 13,800 | 9.9 | 17,424 | 5.6 |
| $TLP_2$ (tonnes) (post-expansion) | 38,052 | 29.5 | 24,963 | 56.6 |
| $\bar{L}_1$ (tonnes) (pre-expansion) | 7,319 | 47.6 | 9,086 | 28.0 |
| $\bar{L}_2$ (tonnes) (post-expansion) | 19,206 | 44.4 | 11,216 | 28.6 |

in the post-expansion period (Table 2). Concurrent with higher biomass, the stock enters into fluctuations after 2011. In both stocks, fishery removals are below start-of-season-biomass and generally below the average total latent productivity (TLP, Fig 4).

Parameter estimates and derived biological reference points from surplus production models selected for each species are shown in Table 2. *O. maya* has a higher biomass ($K$) and lower productivity ($r$) than *O. americanus*. In both species and due to fluctuating dynamics the MSY is an excessive harvest rate, with the notable exception of *O. americanus* in the pre-expansion period when biomass stability permitted valid estimation of the MSY, which is actually close to the average TLP for that period. In general the models for both species are estimated with very good statistical precision. Sustainable harvest rates in both, the pre- and post-expansion periods, are substantially higher than corresponding average landings, although landings of *O. maya* were at the level of the TLP in 2018 and 2021 (Fig 4).

Stock biomass projections show that implementation of the MSY as harvest rate would lead to stock collapse in both species, after the 10 year projections in *O. maya* and during the 10 year projection in *O. americanus* (Fig 5). Conversely, implementation of the average TLP as annual harvest rate would lead to stable biomass in *O. maya* and to nearly stable though declining biomass in *O. americanus*. Furthermore, the projection of *O. maya* biomass under the TLP policy has low uncertainty while the projection of *O. americanus* biomass is much more uncertain, with non-negligible probability to lead to stock collapse, a consequence of the higher statistical uncertainty in estimation of parameters for *O. americanus* (Table 2). The other landings scenarios presented in Fig 5 lead to stable though somewhat more fluctuating and more uncertain biomass in *O. maya*, and steeper declines and high uncertainty in *O. americanus*.

## Discussion

Both assessed octopus stocks in the Yucatan Peninsula showed fluctuating dynamics, *O. maya* over the entire time series and *O. americanus* since 2011. This latter example serves as an

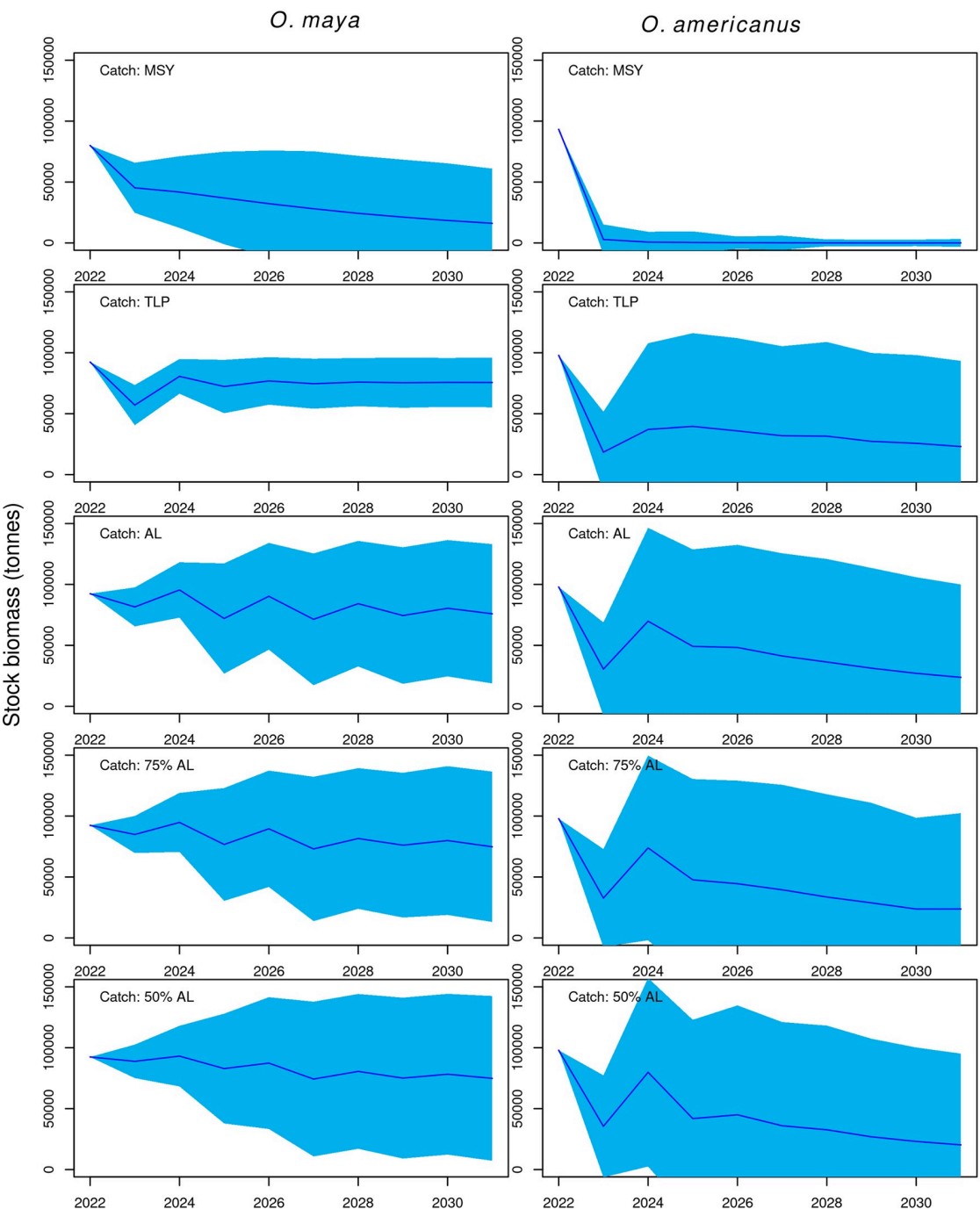

**Fig 5. Projections.** Stochastic biomass projections of the stock of *O. maya* and *O. americanus* in the Yucatan Peninsula 10 years into the future under five landings scenarios. Uncertainty bands (light blue polygons) correspond to two standard errors above and below the mean (blue lines) over 1000 replicates of the projections. These projections are implemented with parameters and standard errors of the selected Pella-Tomlinson model of each octopus species (Table 2). MSY: maximum sustainable yield; TLP: average total latent productivity; AL: average landings over the period 2000 to 2021.

empirical demonstration of the main argument that we present in this work, that the average total latent productivity (TLP) should replace the MSY for stocks that have oscillating or unstable dynamics. During the period of biomass stability of *O. americanus* (2000 to 2010) the MSY was a temporarily valid measure of sustainable harvest rate, estimated at 20 thousand tonnes, close to the average TLP estimate of 17 thousand tonnes. In the second period (2011 to 2021), characterised by a much higher carrying capacity *K*, the biomass of *O. americanus* experienced wide fluctuations and the MSY became an invalid and excessive estimate of surplus production, while the average TLP raised to a reasonable 25 thousand tonnes. In fact, a policy based on harvesting the MSY would collapse the stock of *O. americanus* in less than 10 years and the stock of *O. maya* over a longer horizon with high probability.

Our results are relevant for the management of numerous fish and invertebrate stocks under fishery exploitation, particularly short-lived species, but also long-lived species with special biological characteristics [10–16]. The MSY should be replaced as a limit harvest rate and as a generator of limit biological reference points such as the $B_{MSY}$ and $F_{MSY}$, for all fish and invertebrate stocks that are suspected to have high intrinsic rates of population growth, *r*. This may include all cephalopods, small pelagic fish, some large pelagic fish with short life and fast dynamics such as the dolphinfish [73], and some benthic invertebrate species [15, 16]. Probably a useful classification criterion could be based on a range of *r* provided the model fitted has the parameterization in Eq 3. Stocks that have *r* lower than 1.5 could be managed by recourse to the MSY, stocks with *r* between 1.5 and 2 are in the grey zone, while stocks with *r* higher than 2 should be managed by recourse to the average TLP. These policies for sustainable fisheries should be incorporated in official fisheries management policies at the national and international levels.

In *O. maya* instabilities were apparent during the entire time series and these fluctuations amplified in some periods (2004 to 2008 and 2014 to 2016, Fig 4). Longer instabilities in the pre-expansion period led to more excessive estimation of the MSY for that period whereas the average TLP is still high compared to landings though much lower than the MSY, which further re-enforces our main argument. The general relevance of this argument depends on at least two considerations. Firstly, if the MSY is not valid as a sustainable rate and generator of reference points for fluctuating stocks, is average TLP a valid estimate?, and secondly, how common are fluctuating stocks in the set of fished stocks?

Regarding the first consideration, while the MSY is an invalid sustainable harvest rate or generator of reference points for fluctuating stocks by mathematical necessity, there is no comparable mathematical determination of the average TLP as the valid sustainable harvest rate or generator of reference points for these stocks. Nevertheless, the average TLP makes sense as a replacement of the MSY for fluctuating stocks for several reasons. First, the latent productivity is the general production concept of which the MSY is a particular case [25]. Second, the TLP is not a constant, as is the MSY, but instead it depends on stock biomass and thus it varies year to year taking into account the magnitude of biomass fluctuations. Third, the average TLP, averaged over years, can be interpreted as the basal surplus production that varies annually below the troughs of biomass fluctuations, thus it always returns harvest rates that leave a positive amount of escapement biomass after the fishing season. Note that in fluctuating stocks, population biomass and latent productivity have out of phase cycles: when biomass is high latent productivity is negative (i.e. instead of surplus production there is deficit production) and conversely when biomass is low productivity is positive. This inverse connection between biomass and productivity is ultimately caused by density-dependence. The average TLP is the net balance of surpluses and deficits, and this net balance has to be positive in a viable fluctuating population and a sustainable fishery. A final point in favour of the average TLP as a replacement of the MSY for fluctuating stocks is that unlike the MSY, which is never observed,

the average TLP has a latent and an observed part: in Eq 6 the first term on the r.h.s. is latent and estimated while the second term, the total annual catch, is observed.

Fisheries management has a wider set of tools than just the MSY and reference points derived from the MSY [74]. Nonetheless, the MSY plays a very prominent role in international and national fisheries legislation. So the second consideration concerns the issue of what proportion of all fished stocks are expected to have fluctuating dynamics and therefore how prevalent is the need to have an alternative to the MSY to estimate sustainable harvest rates or to generate reference points. Recently, we have observed fluctuating dynamics in short-lived and long-lived life history types, namely squids [46], sea urchins [16], lobsters [15, 56], and octopus [20]. Fluctuating population dynamics has been an important research topic in fishery science. Beddington and May [75] analysed simple Graham-Schaefer type surplus production models, which posses a stable equilibrium point, and noted that the time to return to the equilibrium point after a disturbance increased as the fishing harvest rate approaches the MSY. So even in populations governed by stable dynamics random disturbances may keep the stock fluctuating for possibly extended periods of time. Mullon *et al.* [76] analysed 1,519 landings time series in FAO database spanning 50 years that included 366 collapses finding that one of three patterns of collapse consisted in sudden drops to zero landings after years of apparent landings stability. The authors interpreted these collapses as a consequence of hidden raises in fishing effort and depensatory mechanisms at low population size. Hsieh *et al.* [77] and Anderson *et al.* [78] studied decadal-long time series of California Cooperative Oceanic Fisheries Investigations records of fish larvae finding that age- and size-truncation of fished stocks increased instability in dynamics by changing demographic parameters such as *r*. Glaser et al. [9] analysed over 200 time series of indices of abundance from surveys and landings from two distinct ecosystems to conclude that instabilities in populations governed by nonlinear models and harvested by fishers make predictive capacity extremely limited in fisheries systems. Essington *et al.* [79] analysed time series of biomass of forage fish covering nearly two thirds of the global forage fish landings finding that naturally occurring biomass fluctuations can be substantially amplified under fishing pressure and lagged responses of the stocks to productivity reductions. In this connection, Hilborn *et al.* [13] noted that forage fish are highly fluctuating stocks making estimation of MSY and derived reference point $B_{MSY}$ quite uncertain. These studies show that fluctuating dynamics is fairly common among harvested fish and invertebrate stocks and therefore our point that the average TLP should replace the MSY as the focus of fishery management for fluctuating stocks is relevant, for all fisheries whose biological reference points derive from the MSY.

Changes in the dynamics of *O. maya* and *O. americanus* around the Yucatan Peninsula have been discovered in this study. The biomass of *O. americanus* increased substantially in the second decade of our study period, during the 2010s. It is rather uncommon that wild fish and invertebrate stocks that are harvested constantly for decades increase in biomass. Based on indirect evidence it has been hypothesized that octopus and more generally cephalopods populations have been proliferating worldwide [30]. In the Yucatan Peninsula, Arreguín-Sánchez [80] has argued that global warming trends and the fisheries depletion of finfish that are competitors or predators of octopus have created more favourable environmental conditions for octopus. Incidence of the upwelling derived from the Loop Current, present during spring and summer, cools the bottom water to 20˚C throughout the sampling area [34], which would favour the events of population aggregations and reproduction, as shown at least for *O. maya* [59], which tends to present a uniform population abundance along the coastline of the Campeche Bank [81] and a reported preference for cold temperate waters [82, 83]. We did not analyse environmental variables or the abundance of finfish during our study period so an environmentally-driven expansion of *O. americanus* around the Yucatan Peninsula cannot be ruled out.

Further clues pointing to an alternative explanation for the larger biomass of the *O. americanus* stock in recent years come from governmental initiatives to help the octopus fishery in Yucatan. Large subsidies in oil expenses and fleet renovations were allocated from the government budget to the fleet in 2011 and later years [84]. This led to fishers increasing the number of named localities that turned into new fishing grounds as reported in the *Avisos de Arribo* database (60% increase in the post-expansion period, 2011–2021, with respect to the former period, 2000 to 2010) and increased spatial coverage of fishing events as recorded in the VMS database (Fig 6). Thus, it seems that at least part of the explanation for the larger biomass in recent years is that fishers simply covered more areas, either new fishing grounds inside the main region of operations or over enlarged regions, that added new stock biomass into the vulnerable fishing biomass. This implies that biomass stability in the pre-expansion period in *O. americanus* probably occurred because of fishing in the core population area of the stock,

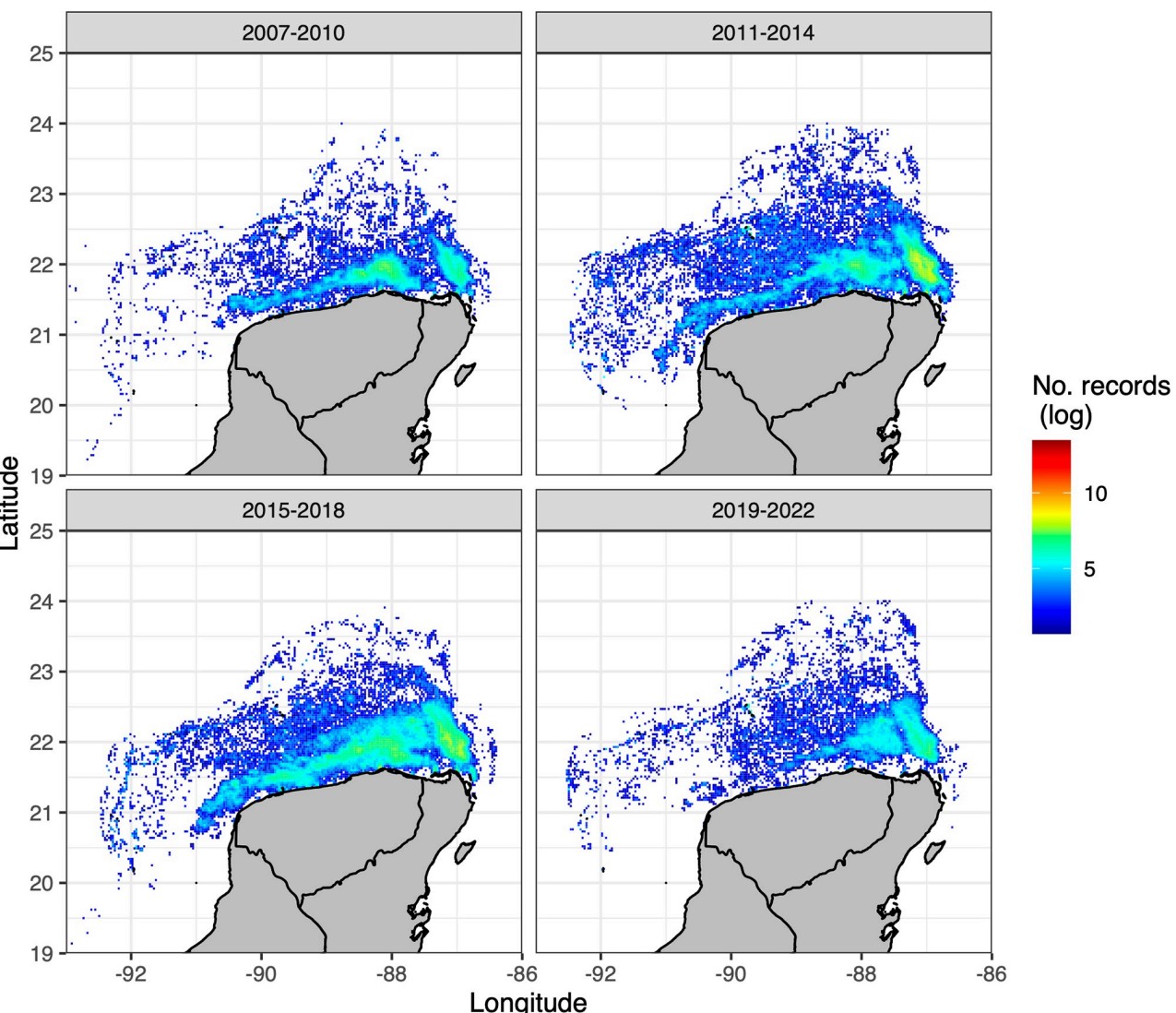

**Fig 6. Spatial distribution of fishing effort.** Spatial coverage of octopus fishing events over four periods in the VMS database for octopus fishing in the Yucatan Peninsula. The top left panel shows the coverage in the pre-expansion period while the other three panels show the coverage during the post-expansion period.

while in the post-expansion period they additionally operated over the wider margins of the stock, including variable borders whose periodic shrinking and expansion may underlie observed fluctuations in total biomass (Fig 4). These changes in the fishing tactics of the fleet have consequences in the estimation of parameters of the model for *O. americanus* in the second stage of our methodology. Interestingly, it appears that these changes in the fleet only affected the estimate of the carrying capacity of the environment *K*, which was four times higher in the post-expansion period (Table 2). This result is consistent with the interpretation above that before the expansion of the fishing grounds, the fleet was fishing in the core of the distribution of *O. americanus* and later expanded to a large outer periphery.

The drop in natural mortality in *O. maya* in the most recent years suggests that there might indeed be more favourable environmental conditions for this species in the Yucatan Peninsula area in recent years, as argued by Arreguín-Sánchez [80] and Ángeles-Gonzalez *et al.* [85]. Two considerations favour the hypothesis of more favourable environmental conditions in recent times [30, 80]. First, Arreguín-Sánchez [80] found statistically significant connections between *O. maya* landings and sea surface temperature and irradiance, interpreting these findings as mediated by stronger octopus somatic growth under higher temperatures. This finding however is counterbalanced by known benefits of colder upwelled waters (less than 27ºC) on *O. maya* physiology [86], sexual maturation [87] and embryo hatchling performance [88]. Probably a balance of higher overall temperatures and upwelling cooling constitute favourable environmental condition for octopus. Second, Arreguín-Sánchez [80] also showed statistically significant inverse connections between octopus landings and red grouper landings with the latter being a predator of octopus [89]. Grouper landings in Yucatan have decreased nearly 50% since 2015. Therefore, we interpret our results of lower natural mortality in *O. maya* in the Yucatan Peninsula as evidence of more favourable environmental conditions for this species in the region.

We found large inter-annual variability in natural mortality rate *M* in both species. This variability in *M* was not an artifact of statistical uncertainty as the maxima and minima of *M* in both species were estimated with reasonably good standard errors. The variability was larger in *O. maya* than in *O. americanus* and although in the former species part of this variability can be explained by decreasing *M* in recent years, as discussed above, the range of values was still very large. For instance, *M* was very high in *O. maya* in 2015, well into the period of decreasing *M*. We hypothesise that this range of variation is fundamentally due to very strong density-dependence leading to strong cannibalism. Two considerations support this hypothesis. First, as can be observed in Fig 2, the highest mortality rate in 2005 in both species coincided with the lowest recruitment estimates. Thus it seems that natural mortality affected the new cohort settling in the fishing grounds. Second, such large inter-annual variability in *M* has also been observed in *O. vulgaris* in Asturias, northern Spain [20], spanning one order of magnitude. In that species it was discovered that the spawning stock and recruitment relationship was best described by the Shepherd model. This spawning stock and recruitment relationship has even stronger density-dependence than the Ricker model. We cannot test this hypothesis in the present case because the fishing season in Yucatan does not include the main and complete spawning period, as is the case with the Asturias octopus fishery, so the fishing does not produce relevant observations. Fishery independent data on the magnitude of spawning in one year and the magnitude of recruitment next year would have to be collected to test the hypothesis that the cause of large inter-annual variability in *M* in these two species of octopus is caused by very strong density-dependence.

Our results agree well with previous assessments of the *O. maya* stock around the Yucatan Peninsula. Jurado-Molina [90] used a landing time series extending from 1995 to 2008 and an index of abundance from surveys extending from 2002 to 2008, to fit a Graham-Schaefer

surplus production model under Bayesian inference. Although Jurado-Molina's [90]$r$ estimate is too low on account of an informative prior distribution, his estimates of the carrying capacity of the environment (with uninformative priors) and sustainable harvest rates are close to our estimate for the pre-expansion period.

The stocks of *O. maya* and *O. americanus* in Yucatan appear not to be overfished or undergoing overfishing although *O. americanus* is experiencing wider fluctuations in biomass in recent years. Instantaneous exploitation rates have been close to 40% since 2017 (except 2019) and realised harvest rates have been below the average TLP. Management measures such as mandatory use of a highly selective fishing gear and closing the fishing season for half of the year are certainly contributing to sustainability, but further measures to avoid too wide fluctuations in biomass might be necessary for *O. americanus*. According to the hypothesis that octopus stocks have unstable equilibrium points [20] and that fishing pushes the stock out of the unstable equilibrium and into cyclic fluctuations, excessive fishing effort and landings may cause excessive fluctuations, thus increasing the risk of undesirable outcomes. Policies that establish well enforceable catch limits could consider the estimate of average TLP in Table 2 for the current expansion period minus an offset due to statistical uncertainty, as it is done in Asturias, northern Spain [12]. Establishing the upper catch limit of each species as the estimated TLP for each species would have positive ecological and economic consequences. From a population dynamics point of view, Fig 4 shows that estimated TLPs minus an offset due to statistical uncertainty would imply removals that are below the low points in biomass fluctuations, which might be termed navigating under fluctuations, in order to avoid applying excessive rates when the stock is passing through low biomass states. In years when the stocks are passing through the high biomass states, the excessive production would just remain in the ecosystem, thus contributing to higher levels of transfers through trophic chains. The recommended limit harvest rates would also keep total landings at the level of historical landings, which places this at the third largest octopus fishery in the world, thus securing the supply of markets that provide for significant revenue to Yucatan fishers' communities. In addition, fishers would be supplying their markets with greater expected stability, since the recommended limit harvest rates are demonstrated here to be sustainable in the long term. Expected stability of supply on the other hand, would improve fishers' bargaining power vis-à-vis purchasers of their output.

Our results were obtained using a specific two-stages stock assessment methodology based on a non-Bayesian hierarchical statistical inference framework. This methodology has been presented earlier and applied to several cases [12, 16, 55], including with time-varying parameters [54], but still is a relatively new methodology so it is worth summarizing here its main characteristics and limitations. The hierarchical two-stages methodology is a stock assessment method for data-limited fisheries. These are the large majority of fisheries worldwide lacking in biological laboratory data such as ageing and fishery independent data such as direct observations from scientific research surveys. The main limitation of our method is that it requires fishing data aggregated at high temporal resolution, either the day, the week or the month as the coarsest possible resolution. All fisheries that have data aggregated to the year are outside the scope of our method. In the applications in this paper the data could be aggregated to the week so each season was analyzed separately while with monthly data all seasons are analyzed simultaneously [54]. These high time-resolution data, although highly informative for parameters that are difficult to estimate inside a stock assessment, in particular natural mortality, contain numerous sources of noise that operate at rapid time scales. This noise may explain the large standard errors in several recruitment estimates (Fig 3). The second limitation having the largest impact affects the depletion model at the first stage. This is that effort and catch must have a positive and strong statistical connection. Thus, in a

simple plot of effort as predictor and catch as response, the cloud of points should be ascending and rather tightly packed around an expected linear or power curve. A wide scattering of points in this effort-catch relationship would cause imprecise estimates of parameters in the depletion model. This imprecision will be passed to the surplus production model at the second stage, leading to large statistical uncertainty about stock status and productivity, or worse, failure to converge entirely. In our study case this potential issues did not occur and on the contrary, estimates of parameters of the surplus production model, and consequently of stock biomass, were very precise (Table 2), except for the average TLP in the recent, post-expansion period. The precision in most parameters of the surplus production model is the result of the hierarchical inference framework (Eq 7) which gives more weight to more precise estimates of stock biomass coming from the depletion model, as well as to the fact that the best variants of the surplus production model had just four parameters to be estimated for each species with 22 years of biomass and landings observations. Other applications of this methodology to long time series of data have also produced high levels of statistical precision [16, 54] while applications to short time series yield imprecise estimates [15]. The third major limitation is not exclusive to our method, as it affects nearly all stock assessment methods and it concerns the surplus production model at the second stage: the length of the time series in terms of number of years must be sufficient to estimate parameters of population dynamics. The shortest time series used to assess a stock with our methodology was five years of monthly effort and catch data of the pink spiny lobster in Mauritania [15], although in that case the data for the surplus production model was augmented with eight additional total annual landings.

Implementation of our recommended policy for the fisheries management of fluctuating stocks in multilateral and national fisheries management policies would require moderate change in stated goals, data collection and data analysis. First, goals need to be generalized from the current focus on the MSY to the more general concept of surplus production called the TLP, of which the MSY is a special case. For the large class of fish and invertebrates with stable dynamics, i.e. those that descend smoothly, quasi-linearly to a lower equilibrium once they experience fishing removals, the TLP is a constant that equals the MSY so the biomass that produces the MSY and the fishing mortality induced by removing the MSY are adequate limit reference points. On the other hand, for the other large class of stocks that enter into fluctuations after fishing removals (or even before any fishing removal), TLP is not a constant and the MSY does not exist. Nevertheless, those stocks still produce surplus production although in them it varies following fluctuations in biomass. Thus two options for policies of sustainable harvesting appear: variable harvest rates that follow the fluctuations in surplus production, or more or less constant harvest rates that average away surplus production fluctuations. We have implemented here the second option (the *average* TLP, where the averaging happens over many years) to sustainable harvest rates of two fluctuating stocks because it seems to us that variable harvest rates that follow the cycle of biomass fluctuations (i) are difficult to implement and enforce not just in our study case but generally, and (ii) would lead to fluctuating supply to fishers' markets which is undesirable from economic points of view. Second, data collection systems in data-limited fisheries would need to be improved to build high time resolution databases of total catch and total fishing effort, at least to the level of the month but preferably to the level of the week. In many data-limited fisheries, total catch and total effort data are estimated by raising from samples of observed fishing trips. In those cases, the sample of observed fishing trips would need to be large enough to have the capacity to estimate daily, weekly or at worst, monthly total catch and effort. In data-rich fisheries, data collection systems are already conducive to management policies based on the TLP. Third, data analysis for management oriented to the TLP would require trivial extensions because the same tools used to estimate

MSY, the biomass at MSY and the fishing mortality, are used to estimate the average TLP, with one exception: the Kobe plot. This phase diagram has become a widely used tool to deliver status assessment of fished stocks [13, 91, 92] because it is well understood by scientists, managers and fishers. Unfortunately, the current Kobe plot is only valid for stocks with stable dynamics because it depends on fixed reference points connected to the MSY, namely the biomass at MSY and the fishing mortality at MSY, which do not exist for stocks with fluctuating dynamics. How to build an equivalent phase diagram for fluctuating stocks is beyond the scope of this work but we note that this is a task that will need to be undertaken in order to deliver status assessments to stakeholders of stocks with fluctuating dynamics. In the case of the Yucatan octopus fishery, official documents of the Mexican government consider the *O. maya* stock in Yucatan to be in the red segment of the Kobe plot since 2018 [93], but this status was obtained from the MSY-based Kobe plot which we are showing here to be inadequate for both octopus species fished in Yucatan.

## Supporting information

**S1 File. S1 to S22 Figs best intra-annual generalized models—*O. maya*.** Model fit to data (top panel; dots: data; line: model) and residual diagnostics (three bottom panels; left: residual histogram; centre: residual cloud; right: quantile-quantile plot) for 22 fishing seasons of *O. maya* in Yucatan, Mexico.
(ZIP)

**S2 File. S23 to S44 Figs best intra-annual generalized depletion models—*O. americanus*.** Model fit to data (top panel; dots: data; line: model) and residual diagnostics (three bottom panels; left: residual histogram; centre: residual cloud; right: quantile-quantile plot) for 22 fishing seasons of *O. americanus* in Yucatan, Mexico.
(ZIP)

**S1 Table. Full set of intra-annual generalized depletion models.** Total number of variants, number of converged fits and properties of selected fits, in the fitting of intra-annual generalized depletion models to weekly catch, effort and mean weight of organisms in the octopus fishery of the Yucatan Peninsula, Mexico. R1 to R5 are abundance input pulses.
(PDF)

## Acknowledgments

We are thankful to members of the Mexico Yucatan Octopus FIP—drift, rod and line (FIP number: 10470) for coordinating and promoting the collaboration among the researchers and institutions participating in this research. We also thank Concepción Enciso-Enciso for conducting a constructive review of a preliminary version of the first technical report that helped greatly to clarify some issues regarding the data. Inés López-Ercilla and Alexander Arkhipkin read an earlier version of this manuscript and offered substantive editorial comments, for which we are grateful. Dr. Chrispine Nyamweya and one anonymous reviewer provided substantial comments to the first submitted version of the manuscript that greatly improved the presentation of our results. Francisco Vergara, from the Marine Stewardship Council, obtained published documents and articles concerning the octopus fishery in Yucatan. Finally, we want to acknowledge the support of Francisco Fernández Rivera-Melo in obtaining the Avisos de Arribo database from 2000 to 2019. We appreciate IMIPAS support for the benthic resources research program, especially for the octopus fishery in Yucalpetén and Lerma.

## Author Contributions

**Conceptualization:** Alicia Poot-Salazar, Iván Velázquez-Abunader, Polo Barajas-Girón, Rubén H. Roa-Ureta.

**Data curation:** Rubén H. Roa-Ureta.

**Formal analysis:** Alicia Poot-Salazar, Iván Velázquez-Abunader, Rubén H. Roa-Ureta.

**Funding acquisition:** Alicia Poot-Salazar, Polo Barajas-Girón.

**Investigation:** Rubén H. Roa-Ureta.

**Methodology:** Rubén H. Roa-Ureta.

**Supervision:** Alicia Poot-Salazar.

**Writing – original draft:** Iván Velázquez-Abunader, Otilio Avendaño, Polo Barajas-Girón, Saul Pensamiento-Villarauz, Jesús M. Soto-Vázquez, José F. Chávez-Villegas, Rubén H. Roa-Ureta.

**Writing – review & editing:** Iván Velázquez-Abunader, Otilio Avendaño, Polo Barajas-Girón, Ramon Isaac Rojas-González, Jesús M. Soto-Vázquez, José F. Chávez-Villegas, Rubén H. Roa-Ureta.

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
