## [Decision Letter · Decision Letter 0]

8 Apr 2024

PONE-D-24-05947Sustainable fishing harvest rates for oscillating fish and invertebrate stocksPLOS ONE

Dear Dr. Velázquez-Abunader,

Thank you for submitting your manuscript to PLOS ONE. After careful consideration, we feel that it has merit but does not fully meet PLOS ONE’s publication criteria as it currently stands. Therefore, we invite you to submit a revised version of the manuscript that addresses the points raised during the review process.

We look forward to receiving your revised manuscript.

Kind regards,

Abdul Azeez Pokkathappada, Ph.D.

Academic Editor

PLOS ONE

Journal Requirements:

3. We note that [Figure 1] in your submission contain [map/satellite] images which may be copyrighted. All PLOS content is published under the Creative Commons Attribution License (CC BY 4.0), which means that the manuscript, images, and Supporting Information files will be freely available online, and any third party is permitted to access, download, copy, distribute, and use these materials in any way, even commercially, with proper attribution. For these reasons, we cannot publish previously copyrighted maps or satellite images created using proprietary data, such as Google software (Google Maps, Street View, and Earth). For more information, see our copyright guidelines: http://journals.plos.org/plosone/s/licenses-and-copyright.

Reviewers' comments:

Reviewer's Responses to Questions

**Comments to the Author**

1. Is the manuscript technically sound, and do the data support the conclusions?

Reviewer #1: Yes

Reviewer #2: Partly

2. Has the statistical analysis been performed appropriately and rigorously? 

Reviewer #1: Yes

Reviewer #2: Yes

3. Have the authors made all data underlying the findings in their manuscript fully available?

Reviewer #1: No

Reviewer #2: Yes

4. Is the manuscript presented in an intelligible fashion and written in standard English?

Reviewer #1: Yes

Reviewer #2: No

5. Review Comments to the Author

Reviewer #1: General comment:

The paper "Sustainable fishing harvest rates for oscillating fish and invertebrate stocks" presents a comprehensive study addressing a critical gap in the management of fish and invertebrate populations subject to fishing. The authors effectively challenge the traditional use of Maximum Sustainable Yield (MSY) as a one-size-fits-all concept for determining harvest rates, by illustrating its inadequacy for populations exhibiting oscillating or unstable dynamics. Instead, they propose the use of Total Latent Productivity (TLP) as a more appropriate measure for setting sustainable harvest rates in such cases. This recommendation is substantiated through a detailed analysis of two octopus species in the Yucatan Peninsula, utilizing a 22-year time series dataset and employing a two-stage stock assessment methodology with time-varying parameters.

The paper would benefit from a more detailed discussion of the assumptions underlying the two-stage stock assessment methodology and the potential limitations or biases these assumptions might introduce.

Overall, the paper, paper is well written and has scientific merit.

Specific comments

The introduction effectively sets the stage for the study, but it could further emphasize the novelty of using Total Latent Productivity (TLP) over Maximum Sustainable Yield (MSY) in the context of fisheries management. Specifically, expanding on how TLP addresses the limitations of MSY in managing oscillating stocks could strengthen the paper's premise. For example, a comparison in lines 50-55 could be deepened with recent case studies illustrating the failure of MSY in similar contexts.

line 66: Write “O. vulgaris” on first mention.

Clarity on the selection criteria for the two-stages stock assessment model, especially the rationale behind choosing specific time-varying parameters, would be helpful. Addressing potential biases these choices might introduce (Lines 230-235) could further solidify the method's validity.

Lines 419-432: While the projections under different harvesting scenarios are insightful, incorporating a discussion on the ecological and economic implications of these projections would enhance the reader's understanding of the practical impacts of adopting TLP-guided management practices.

Lines 580-590: Consider addressing potential challenges in implementing TLP in existing fisheries management frameworks, including the need for capacity building and methodological adjustments, could provide a balanced view of the path forward.

Reviewer #2: The manuscript presents an alternative to the MSY concept, average total latent productivity (TLP), as a more appropriate basis for reference points or limits with which to assess and manage fishing of “oscillating” fish stocks. The authors demonstrate the application of average TLP with a stock assessment of two octopus stocks along the coast of the Yucatan peninsula. The methods and results are both comprehensive and rigorously executed and seem appropriate for the authors’ main questions. My main concerns are the prevalence of unclear language and typos throughout the manuscript, and pitfalls of their analysis and results not being addressed (see my General comments as well as detailed comments). No reanalysis is needed, but numerous edits and rewriting parts of the Discussion are needed.

General

Use of “oscillations and unstable”. The use of these terms in the text is confusing. Oscillatory dynamics by definition are stationary (they always revert back to the mean) and regular (fluctuations occur more or less at equal time intervals). As a result, I tend to think of oscillations as stable: however, the authors often group oscillating with unstable dynamics and stocks, and differentiate this group from stable. Clearly defining these terms early on, and double checking their use throughout the manuscript are needed.

Sentences that are long and difficult to read and understand. These occur throughout the manuscript and contribute to apparent lack of clarity in some parts.

The Discussion needs text on shortcomings and pitfalls of the methods used in this paper, as well as unaddressed uncertainties. I point out specific results in line specific comments that I see allude to possible shortcomings (e.g. the extreme variability in annual natural mortality estimates, highly precise biomass estimates, changes in fleet characteristics). In general, there are common uncertainties and violated assumptions associated with models that only use fishery-dependent data, which are mostly unacknowledged in the manuscript.

Detailed comments

Lines 8. Do you mean limit reference points? Or limiting stock biomass? This wording (e.g. “limit stock biomass”) is unclear to me. Please rephrase.

Lines 8-11: With regards to conventional manuscript structure and style, stating the purpose of the study usually comes at the end of the Introduction. Additionally, this sentence does not flow well from the preceding sentences, which outline the widespread use and importance of MSY. I expected the last sentence to introduce well-known challenges and pitfalls that counter the use of MSY (but instead says “TLP should be used instead” without any reasoning yet)

13. Here and elsewhere where “instability” is used, as well as “oscillations”. I understand these concepts from a mathematical perspective, but I expect your audience will be mostly non-mathematicians. Or perhaps I am misinterpreting these terms myself. Either way, I suggest these terms be clearly and concisely defined (here or elsewhere in the Introduction). The next sentence somewhat defines instabilities (starting line 14), but should be rephrased more directly for clarity (e.g. “Instabilities are ….”). Same for oscillations (“Oscillations are….”).

26. I think this should be “Oscillating or unstable…”

36. Using “instabilities” alone is unclear. Please be more specific (e.g. do you mean irregular population fluctuations? Or something else?).

41-42. I am unsure what “behavioural dynamics … is stable” means.

46. Unsure what “viable oscillating or unstable stocks” are. Please clarify.

55-56. This sentence is unclear. I think there are extra words that don’t make sense, or missing words.

56. Typo in “iscillating”.

67. Remove “found” (revised sentence is then: “…that have been connected to…”

77-81. This sentence is difficult to understand. Please rephrase for clarity.

81-85. I do not see how this tests your hypothesis, although this may be because I do not fully understand what is “unstable dynamics”. I think clearly defining unstable (and differentiating from stable and oscillating) will address this confusion and other comments I have (see comment for Line 13).

90. Change “originated” to “originating”.

96. Can simplify “recurrent incidence” to “recurrence”.

100. I am unsure what is a “target resource.” Please clarify.

103. “new technological capabilities” is vague. Recommend being more specific (e.g. is it in the fishing gear? The boats?).

105. “…total production…” of what exactly?

116-119. This sentence is confusing and may be improved by splitting into two sentences. Either way, please revise for clarity.

124. “Those fleets….”: Do you just mean the 2nd fleet? Or both fleets? Please revise for clarity.

133. You can split this sentence at “… in similar abundances [8,9]. Hence, …”. This would also improve the clarity of this sentence.

138. Remove the “,” between “length” and “or”.

148. I do not think this “,” is necessary here.

156. “…adding over…” can be simplified to “…of…”.

158. Unsure what “…on the multiplicative effect of carrying a number of alijos that conduct the fishing” means. Please revise for clarity.

159. Please change “contained” to “contains”. And elsewhere, check for consistency in your use of verbs, specifically whether they are past or present. Throughout the methods, usage has switched between past and present tense, which I think has also contributed to some of the lack of clarity noted so far.

173. “For stock assessment purposes, …” is unclear and leaves the reader guessing as to what purpose (and if there is more than one). I understand the implication (a discrete-time model needs equal time steps), but I assume other readers may not and thus recommend being more specific here.

180. I do not know what is “… an error of time attribution…”. Did you throw out the trips >1 week? Or include them any ways? Please rephrase this sentence.

183-186. This sentence is somewhat confusing as well. I think there are too many ideas trying to be presented at once that don’t necessarily make sense when combined together (e.g. “weekly time step stock assessment employed”, and “allows direct estimation of annually-varying fishery”). Recommend revising.

196. Omitting “…from a model for the progression…” would streamline this sentence while still allowing it to say what it currently says (but a little more clearly).

205. Are you resampling directly from the observations (or errors from the loess fit), OR are you generating new random values from the truncated normal based on the mean and standard deviation from your fit?

219-224. This is another very long sentence that is difficult to read. More generally, I suggest to carefully reread the entire manuscript, identify sentences where 3 or more commas are used and nothing is being listed or presented as steps in a process (e.g. lines 214-216 is listing different time series, and this is fine), and break these sentences down further into multiple sentences.

244. For the variance-covariance matrix, is a symmetric, uncorrelated structure used? It seems this would be the only option if you are only using the estimates of the standard error. Please clarify either way.

261. “Therefore” is unnecessary here.

Eqn 1. Is “Rj” just representing recruitment? If there is no other “abundance inputs”, I would just refer to this term as “recruitment” here and throughout the manuscript. This would make clear to the reader there are no other “abundance inputs” considered (e.g. immigrants). Also, perhaps aggregate the bounds on the terms below the equation to make for easier reading (e.g. “t, k, alpha, beta, M, and Rj > 0”)

282-290. Are there objective criteria for identifying the “most positive residuals” that are “standing-out”?

334. I believe “determined” should be in the present-tense (determine).

400. Typo (cominf grom).

438. I suggest having a short sentence before this one clarifying that 44 variants were selected from the 912 converged models based on the model selection criteria (if this is indeed true).

454-456. This is extraordinary variation in natural mortality for both species; for example, a high M of 6.4 1/yr for O. maya. implies a survival proportion of <0.01. There should be some Discussion text devoted to the biological realism of these parameters (and others), and what this implies about the underlying assumptions of the IAGD model and the resulting absolute biomass estimates.

461. This precision should be considered in the context of the underlying assumptions of the IAGD. This is my first encounter with this specific methodology and I suspect there are some fairly strong assumptions about how catches represent both population levels and population dynamics.

498. Suggest omitting “variant” in front of “surplus production models..”. The word “variant” is singular, while multiple “models” are referred to.

530-531. This is a consequence of the information going in (biomass and SE estimates from the IAGD models, which have a lot of observations being fit). I am generally skeptical of such precision because these low values are not seen in other data-rich stock assessments. Some part of the Discussion should address this, if such text does not already exist.

579. Typo here: “… relevance of this argumentmanner”

579. Unsure what is “W” as I do not recall it being previously defined.

581. Use of “the” in “is the average TLP the valid estimate.” As written, this proposes TLP as a definitive solution (which may or may not be what the authors intend). I suggest replacing with “… TLP a valid estimate”, as there exist alternative reference point estimators (MEY, some proportion of virgin biomass, stochastic MSY from Bordet and Rivest 2014, etc.).

Bordet, C. and Rivest, L.-P. (2014) A stochastic Pella Tomlinson model and its maximum sustainable yield. Journal of Theoretical Biology 360, 46–53

606-636. While it is indeed well established many fish stocks exhibit “unstable” dynamics (whether this implies nonstationary productivity, highly variable recruitment, etc.), I think the conclusion that “…the average TLP should replace the MSY…” is not convincing without some discussion on other proposed or used alternative reference point estimators (as noted in my comment for line 581, among others not mentioned). Additionally, management procedures for most fish stocks use harvest control rules and other tools for computing fishing pressure targets or limits, which MSY reference points may or may not be used to configure. For example, Fmsy may be used to set a target fishing pressure above a pre-specified biomass threshold in a harvest rule. Ideally, I’d rather see this paragraph streamlined/shortened, and another paragraph added discussing average TLP compared/contrasted with other common reference points.

638: The Kobe plot was created much earlier than 2017. Double check references.

654-658. Another long and difficult to read sentence.

667-682: In general, changes in fishing fleet characteristics (whether technology creep, trends in fishing behavior, etc.) undermine biomass estimates in these type models, where only catches and effort are the only sources of information. I was glad to see that fleet changes over time are acknowledged. But, as you imply in the last sentence that changes in fishing fleet coverage “may underlie fluctuations in total biomass”, what does this imply about the validity of the TLP estimates? If biomass is actually a signal of changes in fishing coverage, then the Pella-Tomlinson model parameters will be biased. At the very least, there needs to be more discussion of assumptions and likely violations of those assumptions in both models (and what this implies about the estimates).

722: Is this “Conclusion” subtitle misplaced?

Fig. 2. Please add subplot labels (e.g. a), b), c), etc.) and increase the text size within the plots (the labels for the y-axis). Also, the estimates of the k parameters (catchability) from the IAGD models should be included.

Fig. 3. This plot is difficult to read, and I am unsure what the main takeaway is. A suggested improvement is to remove the x-axis ticks and labels of all the inner plots (and only display them at the bottom). An alternative and better suggestion is to just have one plot for each species (side-by-side) with all years in each plot. Then the lines can be shaded or colored along a gradient representing the progression from 2002-2021. It will be easy to see trends and shifts over time then (e.g. lines become lower and lower as time progresses).

Fig. 4. I think it is worthwhile to show the MSY value(s) for reference.

6. PLOS authors have the option to publish the peer review history of their article (what does this mean?). If published, this will include your full peer review and any attached files.

Reviewer #1: **Yes: **Chrispine Nyamweya

Reviewer #2: No

---

## [Author Response · Author response to Decision Letter 0]

21 Jun 2024

Reply to reviewers (rebuttal letter)

Reviewer #1 – Dr. Chrispine Nyamweya

General comment:

The paper "Sustainable fishing harvest rates for oscillating fish and invertebrate stocks" presents a

comprehensive study addressing a critical gap in the management of fish and invertebrate populations subject to fishing. The authors effectively challenge the traditional use of Maximum Sustainable Yield (MSY) as a one-size-fits-all concept for determining harvest rates, by illustrating its inadequacy for populations exhibiting oscillating or unstable dynamics. Instead, they propose the use of Total Latent Productivity (TLP) as a more appropriate measure for setting sustainable harvest rates in such cases. This recommendation is substantiated through a detailed analysis of two octopus species in the Yucatan Peninsula, utilizing a 22-year time series dataset and employing a two-stage stock assessment methodology with time-varying parameters.

Reply: Thanks Dr. Nyamweya for your assessment of our work.

The paper would benefit from a more detailed discussion of the assumptions underlying the two-stage stock assessment methodology and the potential limitations or biases these assumptions might introduce.

Reply: Agreed. We have added a paragraph to the Discussion explaining the three main limitations of the two-stages stock assessment methodology. See L. 791-833.

Overall, the paper, paper is well written and has scientific merit.

Specific comments

The introduction effectively sets the stage for the study, but it could further emphasize the novelty of using Total Latent Productivity (TLP) over Maximum Sustainable Yield (MSY) in the context of fisheries management. Specifically, expanding on how TLP addresses the limitations of MSY in managing oscillating stocks could strengthen the paper's premise. For example, a comparison in lines 50-55 could be deepened with recent case studies illustrating the failure of MSY in similar contexts.

Reply: Agreed. We have added brief extensions to the contrast of MSY and TLP in the Introduction, in the paragraph suggested by the reviewer, and reference to the failure of MSY for small pelagic fish in a recent review paper. See L. 60-64.

line 66: Write “O. vulgaris” on first mention.

Reply: Since this was the first mention of a species by its scientific name, we understood that the reviewer correctly pointed out that we should write the full scientific name. We have done that.

Clarity on the selection criteria for the two-stages stock assessment model, especially the rationale behind choosing specific time-varying parameters, would be helpful. Addressing potential biases these choices might introduce (Lines 230-235) could further solidify the method's validity.

Reply: We have expanded and re-organized the explanation of model selection in subsection Intra-annual generalized depletion (IAGD) models. See the 4th paeragraph of this subsection. L. 325-354.

Lines 419-432: While the projections under different harvesting scenarios are insightful, incorporating a discussion on the ecological and economic implications of these projections would enhance the reader's understanding of the practical impacts of adopting TLP-guided management practices.

Reply: Agreed, and thanks for this useful comment. We have expanded paragraph 12th of the Discussion to discuss advantages of the recommended TLP-based limit harvest rates from ecological and economic viewpoints. See L. 764-791.

Lines 580-590: Consider addressing potential challenges in implementing TLP in existing fisheries management frameworks, including the need for capacity building and methodological adjustments, could provide a balanced view of the path forward.

Reply: Agreed, and thanks again, this is another interesting extension of our Discussion. We have added a paragraph at the end of the Discussion (par. 14th) to point out three aspects of implementing TLP-based management in existing fisheries. We have also moved the paragraph about a new Kobe plot to this paragraph because it fits into the topic you raised. See L. 835-878.

Reply to reviewers (rebuttal letter)

Reviewer #2 – Anonymous

The manuscript presents an alternative to the MSY concept, average total latent productivity

(TLP), as a more appropriate basis for reference points or limits with which to assess and manage fishing of “oscillating” fish stocks. The authors demonstrate the application of average TLP with a stock assessment of two octopus stocks along the coast of the Yucatan peninsula. The methods and results are both comprehensive and rigorously executed and seem appropriate for the authors’ main questions. My main concerns are the prevalence of unclear language and typos throughout the manuscript, and pitfalls of their analysis and results not being addressed (see my General comments as well as detailed comments). No reanalysis is needed, but numerous edits and rewriting parts of the Discussion are needed.

Reply: We apologize for the many typos and lack of clarity in the original ms. but we thank you for your overall positive evaluation. We also want to express our gratitude for a thorough review both from the editorial and conceptual points of views. Thank you very much.

General

Use of “oscillations and unstable”. The use of these terms in the text is confusing. Oscillatory dynamics by definition are stationary (they always revert back to the mean) and regular (fluctuations occur more or less at equal time intervals). As a result, I tend to think of oscillations as stable: however, the authors often group oscillating with unstable dynamics and stocks, and differentiate this group from stable. Clearly defining these terms early on, and double checking their use throughout the manuscript are needed.

Reply: Agreed. We have replaced the terms “oscillatory” and “unstable” with the more general term “fluctuations” in all sections of the ms. where we refer to the general phemomenon of not staying at a single equilibrium point, including the title. We also now define fluctuations as including both oscillations and instabilities in the first sentence of the Abstract, and in the second paragraph of the Introduction.

Sentences that are long and difficult to read and understand. These occur throughout the manuscript and contribute to apparent lack of clarity in some parts.

Reply: We have searched for too long sentences throughout the ms. splitting them to improve clarity.

The Discussion needs text on shortcomings and pitfalls of the methods used in this paper, as well as

unaddressed uncertainties. I point out specific results in line specific comments that I see allude to possible shortcomings (e.g. the extreme variability in annual natural mortality estimates, highly precise biomass estimates, changes in fleet characteristics). In general, there are common uncertainties and violated assumptions associated with models that only use fishery-dependent data, which are mostly unacknowledged in the manuscript.

Reply: Agreed partially. 

(1) At the request of another reviewer, we added a paragraph to the Discussion explaining three limitations of our methodology. See L. 792-834.

(2) However, we do not consider that highly precise biomass estimates are artificial or a result of violations of assumptions. Biomass estimates from depletion models for O. americanus had a mean CV of 53%, maximum CV of 272% and minimum CV of 8% across the 22 years of estimates, and biomass estimates from O. maya had a mean CV of 23%, maximum CV of 130% and minimum CV of 6% across the 22 years of estimates. Also Fig. 4 shows reasonable error bands for the biomass time series estimated by the Pella-Tomlinson model. One advantage of our methodology is that it has a self-weighting likelihood function, which by virtue of using the standard error of estimation of each biomass estimates, up-weights precise biomass estimates and down-weights imprecise biomass estimates when fitting Pella-Tomlinson’s model to biomass and annual landings data. This feature of the likelihood in eq. 7 permits more precise estimates of the biomass time series from the Pella-Tomlinson model.

Detailed comments

Lines 8. Do you mean limit reference points? Or limiting stock biomass? This wording (e.g. “limit stock biomass”) is unclear to me. Please rephrase.

Reply: Re-phrased. See L. 7-10.

Lines 8-11: With regards to conventional manuscript structure and style, stating the purpose of the study usually comes at the end of the Introduction. Additionally, this sentence does not flow well from the preceding sentences, which outline the widespread use and importance of MSY. I expected the last sentence to introduce well-known challenges and pitfalls that counter the use of MSY (but instead says “TLP should be used instead” without any reasoning yet)

Reply: OK. Some authors want to go straight to the point at the start of the Introduction but our attemtp at doing that did not yield good results. So we deleted this sentence. But we do not want to start pointing out to all other criticisms of the MSY because we feel that those are largely resolved by using the MSY as a generator of limit reference point, instead of a desired harvest rate.

13. Here and elsewhere where “instability” is used, as well as “oscillations”. I understand these concepts from a mathematical perspective, but I expect your audience will be mostly non-mathematicians. Or perhaps I am misinterpreting these terms myself. Either way, I suggest these terms be clearly and concisely defined (here or elsewhere in the Introduction). The next sentence somewhat defines instabilities (starting line 14), but should be rephrased more directly for clarity (e.g. “Instabilities are ….”). Same for oscillations (“Oscillations are….”).

Reply: Agreed. We have now more precise definitions and have introduced the term “fluctuations” to refer to both, oscillations and irregular instabilities. See L. 11-19.

26. I think this should be “Oscillating or unstable…”

Reply: Agreed, but now we use fluctuations to encompass both, oscillations and irregular instabilities. See L. 28.

36. Using “instabilities” alone is unclear. Please be more specific (e.g. do you mean irregular population fluctuations? Or something else?).

Reply: Clarified. See L. 38.

41-42. I am unsure what “behavioural dynamics … is stable” means.

Reply: In the quoted paper (Halloway et al., 2020) authors use the term ‘behavioral dynamics’ to refer to the change in time of the degree of cooperation inside a group, as opposed to population dynamics, that describe the change in time of population size. They find both dynamics are disconnected to argue that obligate cooperative systems should be unstable at the population dynamics level. This is outside the scope of our paper and not really necessary so we feel that it is better to take out the reference.

46. Unsure what “viable oscillating or unstable stocks” are. Please clarify.

Reply: We have re-phrased that part to improve clarity. What we are saying is that fluctuating stocks (either oscillatory or irregularly unstable) have positive growth rate when they are increasing from low biomass to high biomass, and they have negative growth rates when they are decreasing from high biomass to low biomass. In the mid- to long-term positive growth rates have to be higher on average than the absoute value of negative growth rates, because if the opposite happens, stock biomass will collapse to zero. So ‘viable’ means that in the mid- to long-term positive growth rates are larger than (absolute value) negative growth rate so that biomass remains positive. We explained this is a short format in the revised ms. See L. 43-48.

In other words:

In stocks with stable dynamics, what we want is that the growth rate is always positive. 

In fluctuating stocks the equivalent goal is that the mid- to long-tern average growth rate (averaging over positive and negative growth rates) is positive.

55-56. This sentence is unclear. I think there are extra words that don’t make sense, or missing words.

Reply: Re-phrased to clarify. See L. 54-56.

56. Typo in “iscillating”.

Reply: Replaced with ‘fluctuating’, see L. 56.

67. Remove “found” (revised sentence is then: “…that have been connected to…”

Reply: Done, see L. 71.

77-81. This sentence is difficult to understand. Please rephrase for clarity.

Reply: Done, see L. 81-86.

81-85. I do not see how this tests your hypothesis, although this may be because I do not fully understand what is “unstable dynamics”. I think clearly defining unstable (and differentiating from stable and oscillating) will address this confusion and other comments I have (see comment for Line 13).

Reply: See the changed text at the end of Introduction, L. 86-88.

90. Change “originated” to “originating”.

Reply: Done, L. 93.

96. Can simplify “recurrent incidence” to “recurrence”.

Reply: Done, L. 99.

100. I am unsure what is a “target resource.” Please clarify.

Reply: We meant non-bycatch, but it was better to just delete that fragment, see L. 103.

103. “new technological capabilities” is vague. Recommend being more specific (e.g. is it in the fishing gear? The boats?).

Reply: Agreed, see L. 105-108.

105. “…total production…” of what exactly?

Reply: See L. 108.

116-119. This sentence is confusing and may be improved by splitting into two sentences. Either way, please revise for clarity.

Reply: Done, see L. 120-123.

124. “Those fleets….”: Do you just mean the 2nd fleet? Or both fleets? Please revise for clarity.

Reply: Done, see L. 127-129.

133. You can split this sentence at “… in similar abundances [8,9]. Hence, …”. This would also improve the clarity of this sentence.

Reply: Done, see L. 138-139.

138. Remove the “,” between “length” and “or”.

Reply: Done, see L. 143.

148. I do not think this “,” is necessary here.

Reply: Done, see L. 153.

156. “…adding over…” can be simplified to “…of…”.

Reply: Done, see L. 161.

158. Unsure what “…on the multiplicative effect of carrying a number of alijos that conduct the fishing” means. Please revise for clarity.

Reply: Rephrased for clarity, see L. 162-164.

159. Please change “contained” to “contains”. And elsewhere, check for consistency in your use of verbs, specifically whether they are past or present. Throughout the methods, usage has switched between past and present tense, which I think has also contributed to some of the lack of clarity noted so far.

Reply: Done, and throughout the text for M & M we’ve checked for consistency in verb use, keeping/switching to past tense when we describe things we did and to present tense when we described things that still are. See L. 164.

173. “For stock assessment purposes, …” is unclear and leaves the reader guessing as to what purpose (and if there is more than one). I understand the implication (a discrete-time model needs equal time steps), but I assume other readers may not and thus recommend being more specific here.

Reply: Agreed, clarifications added, see L. 178-179.

180. I do not know what is “… an error of time attribution…”. Did you throw out the trips >1 week? Or include them any ways? Please rephrase this sentence.

Reply: All fishing trips were included. It’s just that some trips that lasted more than one week were assumed to happen entirely in the 1st week in order to have a weekly time step for the depletion models. Another option was to group the data at monthly time steps but that decision would have precluded estimation of time-varying parameters in the depletion model and a great loss of degrees of freedom to estimate parameters (due to having many more weeks than months in a fishing season). So we judged that the loss due to erronous time attribution of some fishing trips was minor compared to the loss of aggregating the data at the monthly time step. See clarifications in L. 186-199. 

183-186. This sentence is somewhat confusing as well. I think there are too many ideas trying to be presented at once that don’t necessarily make sense when combined together (e.g. “weekly time step stock assessment employed”, and “allows direct estimation of annually-varying fishery”). Recommend revising.

Reply: The clarified te

---

## [Decision Letter · Decision Letter 1]

12 Jul 2024

Sustainable fishing harvest rates for fluctuating fish and invertebrate stocks

PONE-D-24-05947R1

Dear Dr. Velázquez-Abunader,

We’re pleased to inform you that your manuscript has been judged scientifically suitable for publication and will be formally accepted for publication once it meets all outstanding technical requirements.

Kind regards,

Abdul Azeez Pokkathappada, Ph.D.

Academic Editor

PLOS ONE

Additional Editor Comments (optional):

Dear authors, kindly incorporate minor corrections suggested by reviewers.

Reviewers' comments:

Reviewer's Responses to Questions

**Comments to the Author**

1. If the authors have adequately addressed your comments raised in a previous round of review and you feel that this manuscript is now acceptable for publication, you may indicate that here to bypass the “Comments to the Author” section, enter your conflict of interest statement in the “Confidential to Editor” section, and submit your "Accept" recommendation.

Reviewer #2: All comments have been addressed

2. Is the manuscript technically sound, and do the data support the conclusions?

Reviewer #2: Yes

3. Has the statistical analysis been performed appropriately and rigorously? 

Reviewer #2: Yes

4. Have the authors made all data underlying the findings in their manuscript fully available?

Reviewer #2: Yes

5. Is the manuscript presented in an intelligible fashion and written in standard English?

Reviewer #2: Yes

6. Review Comments to the Author

Reviewer #2: The authors have thoroughly and carefully addressed all reviewer comments and revised the manuscript accordingly. As a result, the writing has substantially improved in both clarity and rigor. The manuscript should be accepted for publication in PLOS ONE. I have some minor comments (mostly editorial again) for which I do not need to the see authors’ responses.

Minor comments

Lines 103-10: Add conjunction ‘while’ and remove ‘since’ – “… since the ancient Mayans, while commercial operations began in 1949 in Campeche and in 1970 in Yucatan and Quintana Roo.”

Lines 141: Typo in “miss-classified” – replace with “misclassified.”

Lines 143-144: Replace ‘of’ with ‘for’ when specifying values for length and weight - “…legal size of capture of 110 mm for mantle length or approximately 450 g for whole body weight, …”

Line 172: Omit ‘the’ within ‘In the light of stock …’

Line 199: Replace ‘… better capture of the fast growth…’ with ‘…better captures the fast growth…’ to be more direct.

Line 616: The difference between “magnitude” and “amplitude” of biomass fluctuations in this sentence is unclear (as they generally mean the same thing, as I read it). Omit either term, or rephrase to clarify.

Line 664: The sentence implies the name “Kobe plot” originated in 2017, but it has been in use since before 2017 (e.g. see Maunder and Aires-da-Silva, 2011 below). Regardless, the timing does not seem an important detail, so I suggest omitting this part of the sentence (so it reads as “The Kobe plot is a widely used tool…”).

Maunder, M. and Aires-da-Silva, A. (2011) Evaluation of the Kobe plot and strategy matrix and their application to tuna in the EPO. Unpublished IATTC Scientific Advisory Committee document SAC-02-11 La Jolla, USA, 14 pp.

Lines 851-856: I think I do not agree with the statement “…fluctuating harvest rates (i) are beyond the capacity of currently existing management systems…” because at least in developed countries’ fisheries, harvest rates (or fishing mortality) already scale/change with biomass below a certain limit or target reference point (the “hockey-stick” rule). I say “I think” because it is more likely that I misunderstand what is being said here, so I still suggest rephrasing for clarity. Either way, yes, it is economically undesirable.

Lines 866-867: See my suggestion for Line 664.

7. PLOS authors have the option to publish the peer review history of their article (what does this mean?). If published, this will include your full peer review and any attached files.

Reviewer #2: No

---

## [Editor Report · Acceptance letter]

24 Jul 2024

PONE-D-24-05947R1 

PLOS ONE

Dear Dr. Velázquez-Abunader, 

I'm pleased to inform you that your manuscript has been deemed suitable for publication in PLOS ONE. Congratulations! Your manuscript is now being handed over to our production team.

Kind regards, 

on behalf of

Dr. Abdul Azeez Pokkathappada 

Academic Editor

PLOS ONE